biomedical engineering/medical physics

wheezing, starling resistor, stethoscope

**Author for correspondence:**
A. L. Gregory
e-mail: alg57@cam.ac.uk

# An experimental investigation to model wheezing in lungs

## A. L. Gregory, A. Agarwal and J. Lasenby

Department of Engineering, University of Cambridge, Trumpington Street, Cambridge CB2 1PZ, UK

 ALG, 0000-0003-0667-5861

A quarter of the world's population experience wheezing. These sounds have been used for diagnosis since the time of the Ebers Papyrus (*ca* 1500 BC). We know that wheezing is a result of the oscillations of the airways that make up the lung. However, the physical mechanisms for the onset of wheezing remain poorly understood, and we do not have a quantitative model to predict when wheezing occurs. We address these issues in this paper. We model the airways of the lungs by a modified Starling resistor in which airflow is driven through thin, stretched elastic tubes. By completing systematic experiments, we find a generalized 'tube law' that describes how the cross-sectional area of the tubes change in response to the transmural pressure difference across them. We find the necessary conditions for the onset of oscillations that represent wheezing and propose a flutter-like instability model for it about a heavily deformed state of the tube. Our findings allow for a predictive tool for wheezing in lungs, which could lead to better diagnosis and treatment of lung diseases.

## 1. Introduction

Lung sounds offer a cheap, non-invasive, non-radioactive source of information on pathology in the upper chest [1], but diagnoses based on these sounds lack specificity [2] and repeatability [3,4]. Wheezing is one of the most common lung sounds [2,5]. It is indicative of a breathing problem, and usually indicates a reduced airflow. An understanding of the physical mechanism responsible for generating wheezing sounds could provide a better causal link between symptoms and disease and help improve the diagnosis and treatment of diseases.

Wheezing sounds can be made during inhalation or expiration and are strongly tonal. It has been shown by [5,6] that the frequencies of wheezing sounds do not change when the density of the gas is changed by replacing nitrogen with helium as the working fluid. This rules out mechanisms based on the resonance of the air cavity [5] because variations in fluid density

**Figure 1.** The geometry of the flexible tubes used to investigate self-excited oscillations. (*a*) Model of an airway in the lung. The measured quantities are, $d$ the unstrained diameter ($a = d/2$ the unstrained radius), $h$ the wall thickness, $l$ the tube length, $p_1$, $p_2$ and $p_e$ the pressures, and $Q$ the flowrate. Not shown is $l_0$, the unstrained tube length. Also of interest are $p_c$ and $A_c$, the pressure and cross-sectional area of the flexible tube at its narrowest section. (*b*) The geometric parameter range spanned by our experiments. The shaded regions represent the typical values of the geometric parameters found in the upper airways of the lung, derived from the model of Horsfield [9] and the work of Hoppin [10].

would cause variations in the speed of sound and hence the air-cavity resonance frequencies. In an excised dog trachea, it was found that when air was sucked through it, under certain conditions, the trachea itself began to oscillate, creating wheezing sounds [7]. These observations provide strong evidence that wheezing is heard when the flexible tubes that make up the airways of the lung, oscillate as air flows through them, as was first suggested by Grotberg & Davis [8].

Understanding self-excited oscillations as fluid flows through flexible tubes has been the subject of many studies, both experimental and theoretical. The experimental set-up of a flexible tube clamped at both ends with a fluid flowing through it is referred to as a Starling resistor (figure 1*a*). Kamm & Pedley [11, p. 179] provide a review of work prior to 1989, giving the following useful conclusion,

> One of the main lessons to be learned is that it is vital for experimentalists to describe, quantitatively, all aspects of their experiment, because no one yet knows that any of them are unimportant, especially during oscillations, and for theoreticians to recognize that just because they have a model that produces something qualitatively similar to what is observed (e.g. an oscillation) it does not mean they have given a mechanical explanation for any particular observation.

The theoreticians being referred to by [11] produce various low order lumped parameter models of flow through flexible tubes (for example [12,13]). As [11] says, these models produced interesting phenomenology, but are unable to provide a physical explanation to predict the conditions required for the onset of oscillations.

More recent work in the field is reviewed by [14–19]. With the exception of [7,20], the experimental work has used water as the working fluid. These experiments have produced an extensive catalogue of behaviour, but the use of water means that the ratio of the density of the fluid to the tube wall is wrong for the lung by a factor of around $10^3$. In water, the inertia of the tube is negligible compared to the inertia of the fluid, but that is not the case when air is the fluid medium as in the lungs. Therefore, experimental investigations with water are not appropriate to study wheezing in lungs. [7] uses air as the working fluid, but the experiments still have a limited applicability to the lungs. The tubes that make up the airways are relatively short compared to their diameter, which is not true in the experiments of [7]. We also need to consider a wider variety of tube lengths, diameters and wall thicknesses, and most significantly, must consider the effect of axial pre-tension, which is present in the airways [10]. [20] uses air as the working fluid as well, however, it investigates the phenomenon of flow limitation, rather than self-excited oscillations. These phenomena are possibly related, as we see that

oscillations generally start shortly after the maximum flowrate is achieved, but this is not fully established. Also, like [7,20] uses tubes that are too long relative to their diameter, and does not consider the effect of axial pre-tension.

Theoretical work has gradually increased the complexity of the models under consideration, starting with a two-dimensional analogue of the Starling resistor. Both linear stability [21–30] and direct numerical simulation [31–36] have been considered. In a similar way to the lumped parameter models, the two-dimensional models produce interesting phenomenology, but cannot be used quantitatively. Theoretical work on the three-dimensional system has progressed a lot since the 1990s. Linear stability analysis has been completed by [37–40], and more recently by [41,42]. In order to be relevant to the lung, a linear stability analysis would need to include the effects of axial pre-tension, consider tubes whose length is relatively short compared to their diameter (a length of four times the diameter or less is typical [9]), include the effects of the inertia of the tube walls (because air is the working fluid in the lung), and linearize about a strongly deformed tube state (this is one of the main observations of our work). [41] had some success predicting the frequency of self-excited oscillations compared to computational fluid dynamics, but neglected wall inertia, considered long tubes, and linearized about an elliptically cylindrical state. Wall inertia has been included in a linear stability analysis by [42], but the tubes considered are still long (the work assumes that the instability has a long wavelength compared to the tube diameter), and once again the tube is not strongly deformed when oscillations start.

The main aim of this paper is to provide an extensive set of experimental results of direct relevance to wheezing in the lung. Significant progress has been made in developing our theoretical understanding of Starling resistors, producing plenty of candidate mechanisms, but as yet quantitative predictions can not be made for the frequencies and flowrates at onset in the specific context of the lung. We hope the results presented in this paper will help to develop a theory for the onset of wheezing. Based on our results, we present an empirical model to predict the onset of oscillations that lead to wheezing. We then propose a phenomenological model for the onset of wheezing.

## 2. The experiment

The airways of the lung are a branching network of flexible tubes (bronchioles) that gradually get shorter and narrower as we move further into the lung [9,43]. We model bronchioles with flexible tubes using the set-up illustrated in figure 1a. The flexible tubes under investigation are defined by their unstrained length $l_0$, diameter $d$ (radius $a = d/2$), and wall thickness $h$. In the airways, the bronchioles are generally in tension [10]. To reproduce this in the experiments, the tubes are axially stretched to a length $l > l_0$. Hence, there are three dimensionless geometric ratios that define the tubes. The values of these ratios for the tubes used in the experiments are shown in figure 1b, along with the typical values found in the upper airways of the human lung. It can be seen that we have covered a parameter regime of direct relevance to the lung. We used rubber as the material for our flexible tubes, whose material properties are: Young's modulus $E = 1$ MPa, Poisson's ratio $v = 0.5$ and tube wall density $\rho_s = 1000$ kg m$^{-3}$. These values are close to those found in the lung [44–47]. The tubes themselves were cut from the ends of rocket balloons, from the company 'Party Time'. We measured the material properties with an Instron 3400 series universal testing system.

Figure 2 shows a schematic of the experimental set-up used to investigate the oscillations of flexible tubes, which is a zoomed out version of figure 1a. Air flows into the system through (1), then through a rotameter (2) used to monitor flowrate. The noise that the rotameter introduces into the flow, and any other noise, is isolated from the flexible tube by the upstream settling chamber (3). Air flows into the upstream clean flow tube (5) section via a shaped inlet (4) that reduces separation. A contraction (6) leads to the flexible tube (7), before an expansion (6') leads to the downstream clean flow tube (5') that exits into the downstream settling chamber (3'). Suction is provided by a fan (8). The downstream settling chamber (3') isolates the flexible tube from the noise from this fan. Experiments were performed in the Acoustics Laboratory of the Department of Engineering at the University of Cambridge. More detail of the experimental rig is given in [48].

During a typical experimental run, a rubber tube is selected (one of the dots in figure 1b), and attached to the rig. The suction fan is slowly ramped up to full power, held at full power, and slowly ramped back down using an electronically controlled voltage. The whole experimental run takes approximately 10 min. The speed at which the suction fan is ramped up to full power is chosen to be slow enough that any quantities (pressure, frequency etc.) measured at the onset of oscillation are

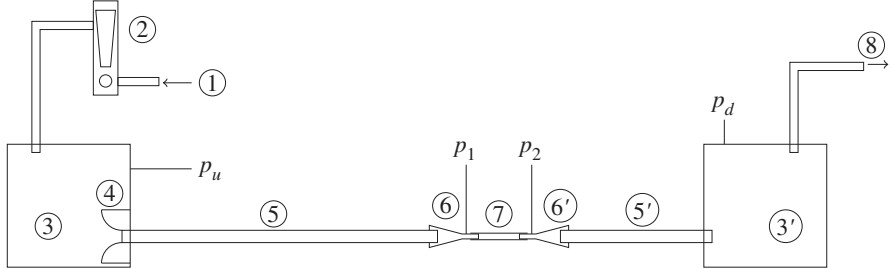

**Figure 2.** Schematic of Starling resistor experiment. 1: flow inlet, 2: rotameter, 3/3′: settling chambers, 4: clean flow inlet, 5/5′: clean flow tubes, 6/6′: contraction and expansion, 7: flexible tube, 8: tube to suction fan. The volume of the downstream settling chamber is approximately 4 m$^3$, and of the upstream settling chamber is 0.03 m$^3$. Pressure is monitored at the four locations indicated, $p_u$, $p_1$, $p_2$ and $p_d$.

independent of the ramping speed, which implies that the flexible tube is in a quasi-steady state at any point during the experimental run.

We measure the hydrodynamic pressure in the upstream ($p_u$) and downstream ($p_d$) settling chambers, just upstream ($p_1$) and downstream ($p_2$) of the flexible tube, and the acoustic pressure ($p_a$) 30 cm outside from the centre of the flexible tube, with a sampling frequency of 51200 Hz using an NI cDAQ-9178, which includes an anti-aliasing filter and logs all of the pressure measurements simultaneously. Hydrodynamic pressures are measured with Kulite XCS-093-5PSID transducers, and acoustic pressure is measured with GRAS 40DD microphones. The pressure sensors give results relative to the external environmental pressure ($p_e$), and from here on all pressures are defined relative to $p_e$. Flowrate is monitored by a Key Instruments MR3000 series rotameter, which is filmed throughout the experiment. The video recording also has an audio stream that is synchronized with the acoustic measurement $p_a$. This means that the timings for the flowrate measurements are synchronized with the other measurements to within 0.1 s.

For each tube in figure 1b at least two experimental runs were completed. The measurements from a single experimental run are summarized in figure 3. Figure 3a shows a moving average of the pressures, $p_1$ and $p_2$, and the flowrate $Q$ as a function of time. Figure 3b shows the spectrogram aligned in time with figure 3a. During the experimental run, suction is increased (the plenum pressure is decreased) between $t_1$ and $t_2$, held constant between $t_2$ and $t_3$, and then decreased between $t_3$ and $t_4$. $\bar{p}_2$ mirrors the suction (plenum) pressure throughout. $\bar{p}_1$ first decreases with increasing suction until just before the time $t_o$, which corresponds to the onset of oscillations. This can be seen from the spectrogram (before $t_o$, there are no tonal sounds, they appear for the first time as noisy events at $t_o$). After this time, $\bar{p}_1$ plateaus out and is largely unaffected by suction pressure until suction is decreased again to a point where the tube stops oscillating. The flowrate behaves in a similar way. It first increases with increasing suction but then drops just before the onset of oscillations and plateaus out after onset. The pink shaded region in figure 3 indicates $\bar{p}_1 - \bar{p}_2$. From this, we can see that the onset of oscillation increases the pressure loss between $p_1$ and $p_2$ and restricts the flowrate.

The data from these runs contain a wealth of information, but here we focus on the onset of oscillations, which is labelled by a vertical dashed line in figure 3. Specifically, we can find the dominant frequency of the oscillations at onset, along with $Q$, $\bar{p}_1$ and $\bar{p}_2$. We take the onset frequency to be the first dominant frequency identified in the spectrograms of $p_1$, $p_2$ and $p_a$ that is sustained for at least 100 cycles (figure 3c).

As well as the experimental runs outlined by figure 3, we have captured stereoscopic high-speed videos of several of the tubes that allowed us to track the real-time motion of the tube at the onset of oscillations (see [49] for details).

In the rig, the air comes from a reservoir at atmospheric pressure (the room), which is equal to the external pressure to the flexible tube, $p_e$. The flow is ultimately sucked into a downstream reservoir at a lower pressure. This set-up was chosen to approximately mimic expiration in the lung. During expiration, air in the lungs is pushed out by the compression of the lung. This compression raises the pressure in the outermost portions of the airways (the parenchyma), which drives flow to the mouth, where pressure is atmospheric. The raised pressure in the parenchyma, which drives the flow, also acts as an external pressure to compress the bronchi and bronchioles. Hence the pressure far upstream and external to the flexible tubes are approximately equal during expiration; a set-up that we have replicated in our rig. We have chosen to focus on expiration because wheezing is much more common during this phase of the breathing cycle [5].

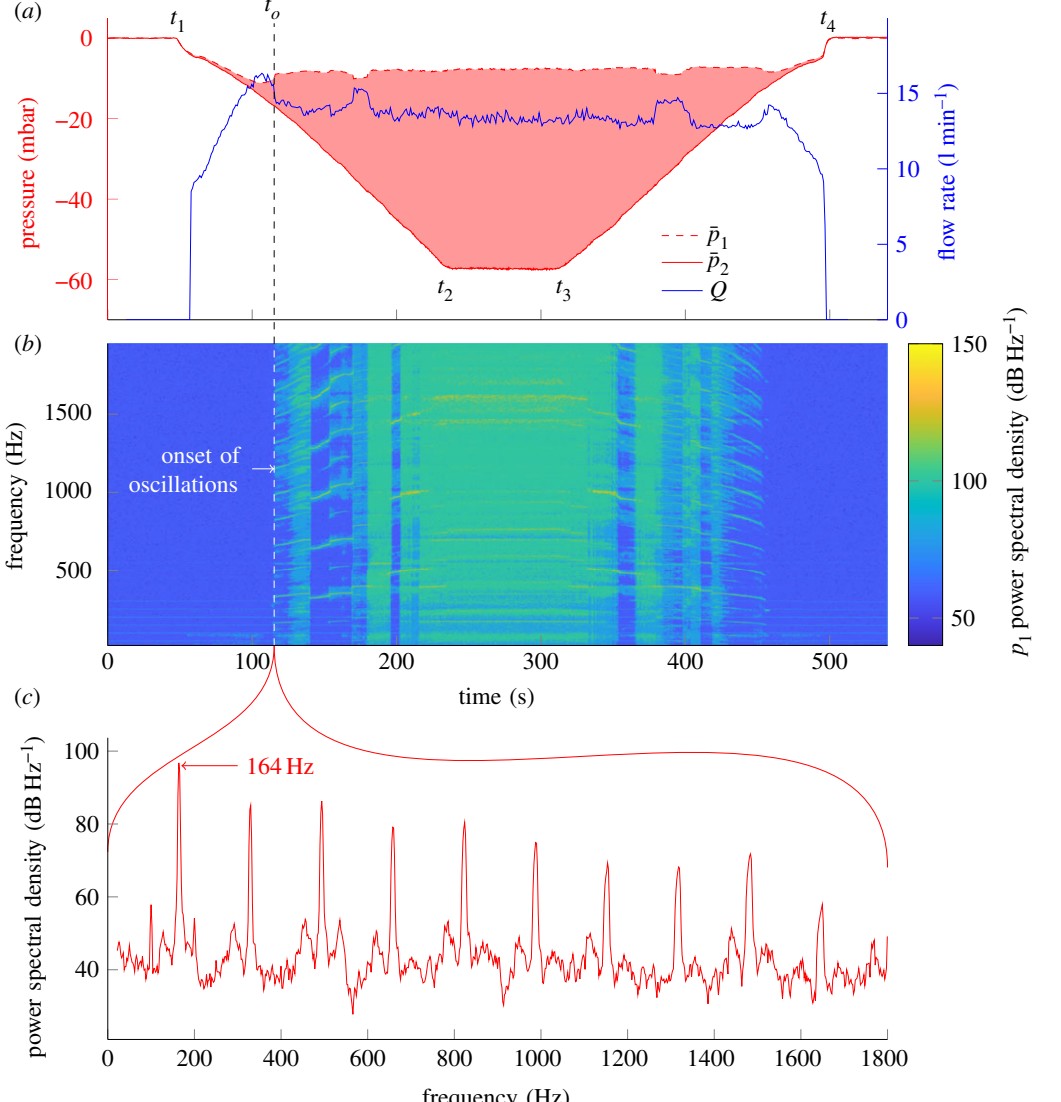

**Figure 3.** Experimental run investigating self-excited oscillations in flexible tubes. The pressure just upstream ($p_1$) and downstream ($p_2$) are measured as the suction ((8) in figure 2) is gradually increased [$t_1,t_2$], held constant [$t_2,t_3$], and then decreased [$t_3,t_4$]. Moving averages of $p_1$, $p_2$ and $Q$ are shown in (a). This plot has been aligned in time with a spectrogram of $p_1$ in (b). The geometric parameters of the tube are $l = 23$ mm, $l_0 = 20.4$ mm, $h = 0.35$ mm, $d = 6$ mm. The power spectral density of $p_1$ is shown in (c) at the onset of oscillations, which indicates the onset frequency (164 Hz).

To test the effect of the length of the upstream and downstream tubes (5 and 5′ in figure 2), we conducted experiments with varying lengths of these tubes but found that it had no effect on the results. The results presented here are for fixed lengths of the tubes.

# 3. High-speed video

High-speed videos of the experimental runs reveal interesting features at the onset of oscillations. Figure 4 shows a set of stills from such a run. The first important observation to make is that the oscillations occur about a strongly deformed state of the tube. The second is that there is a visible wave that travels the length of the tube, and that in the time taken for the wave to travel down the tube, reflect and travel back up to complete one cycle, the tube also opens and closes once. A single phase point of the longitudinal wave has been indicated in figure 4 over the course of one cycle.

The presence of longitudinal waves synchronized with the opening and closing of the tube is strongly indicative that these two oscillation modes couple to give rise to self-excited oscillations. Videos from our study are available in the dataset [50].

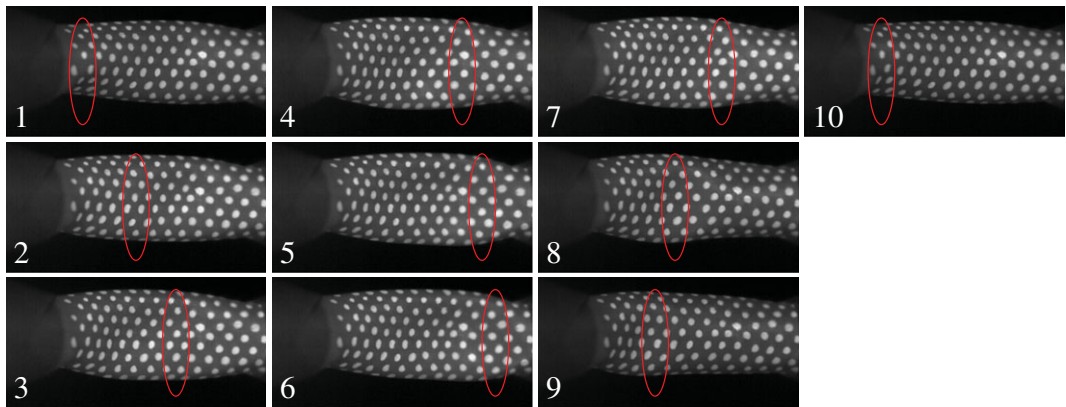

**Figure 4.** High-speed video of self-excited oscillations just after onset. A longitudinal wave is observed, and its position in the sequence of images is indicated by the red ellipse. The elapsed time between each image is 4 ms. We recommend viewing this as a video, the original files are available at [50]. The white dots have been drawn onto the rubber to aid visualization.

To develop our understanding further, we must be able to characterize the behaviour of the system in the lead up to onset in much more detail. Our experiment has been set up to ensure that the experimental rig goes through a sequence of quasi-steady states as onset is approached, and hence the system is moving through a sequence of equilibrium states determined by the fluid and the tube together. Our first modelling step is therefore to characterize the equilibrium behaviour of the tube and fluid separately, as if fluid–structure interactions were absent, before combining these models to understand the system as a whole.

In the absence of flow, the cross-sectional area of the tube at its narrowest point, $A_c$, is a function only of $p_c$ (the pressure at $A_c$), and this function is called the 'tube law'. It describes the equilibrium transmural pressure as a function of area, which we denote $p_c = f_t(A_c)$, where $f_t$ is implicitly dependent on the geometry and material properties of the tube. The tube law describes how the tube contracts and expands as the pressure inside the tube is changed (and then held constant such that there is no flow).

When fluid flows through the tube, $p_c$ is set by the pressure in the fluid at the narrowest cross-section $A_c$. The experimental system is controlled by the level of suction far downstream, i.e. $p_d$. During an experimental run, $p_d$ is decreased slowly so that the system progresses through a sequence of quasi-steady states. To understand the equilibrium of the fluid, we must ask how $p_c$ varies as $A_c$ is changed, assuming that $p_d$ is fixed and known, which is the case in our experimental set-up. Equilibrium of the fluid is governed by $p_c = f_f(A_c, p_d)$, which we call the 'fluid law', where $f_f$ is implicitly dependent on the geometry of the entire rig. Equilibrium of the system as a whole will occur at $A_c = A_{c_0}$ such that $f_t(A_{c_0}) = f_f(A_{c_0}, p_d)$.

## 4. Tube law

Considerable work has been done developing tube laws [51–53], however, we have been unable to find a law that is valid for tubes of finite length held in axial tension that are relevant to the present study. Therefore, we have obtained this law empirically.

We regard $A_c$ as a function of $p_c$, the geometry ($l, l_0, a, h$) and the material properties ($E, v$):

$$A_c = f(p_c, l, l_0, a, h, E, v).$$

We can non-dimensionalize [54] the above relationship for $A_c$:

$$\frac{A_c}{\pi a^2} = f\left(\frac{p_c}{E/(1 - v^2)}, \frac{l}{l_0}, \frac{h}{a}, \frac{l_0}{a}\right).$$

Here, we have assumed that $A_c$ does not depend on $E$ and $v$ independently, but on $E/(1 - v^2)$. This assumption is based on the form taken by the energy of deformation for thin shells, in which $E$ and $v$ only appear in this form [55].

We have measured $A_c/\pi a^2$ as a function of $p_c/(E/(1 - v^2))$ for various values of $l/l_0$, $h/a$ and $l_0/a$, see appendix A for details. We find that if we plot

$$\frac{A_c}{\pi a^2}\left(\frac{l}{l_0}\right)^\alpha \quad \text{against} \quad \frac{p_c}{E/(1 - v^2)}\left(\frac{l}{l_0}\right)^\beta\left(\frac{h}{a}\right)^\gamma\left(\frac{l_0}{a}\right)^\delta,$$

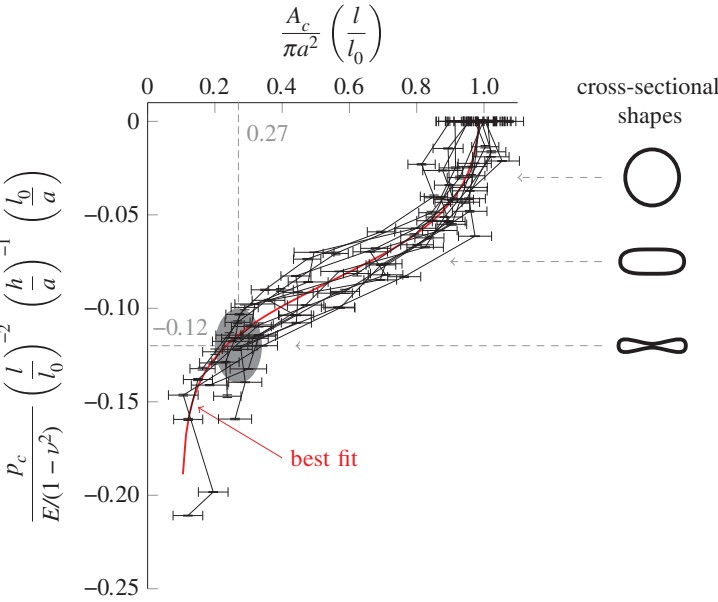

**Figure 5.** An approximate tube law valid for short flexible tubes held under strain. $A_c$ is measured as $p_c$ is reduced for various tube geometries (characterized by $l$, $l_0$, $a$, $h$). By plotting the dimensionless groups shown, these tube laws approximately collapse onto a single curve. $E$ is Young's modulus and $\nu$ is Poisson's ratio. Uncertainty in the measured value of $p_c$, owing to the limitations of the equipment, is represented with vertical error bars. Uncertainty in the measured cross-sectional area, which is obtained through repeat measurements of the area, is represented with horizontal error bars. The shaded region represents the point at which the tube walls come into contact. The cross-sectional shapes of the tube are also sketched at important points.

all our measurements of the tube law collapse onto a single curve. The factor $(l/l_0)^\alpha$ is needed next to $A_c/\pi a^2$ because axially straining the tube reduces the cross-sectional area, even when $p_c = 0$. The optimum exponent values were found to be $\alpha = 1.0$, $\beta = -2.0$, $\gamma = -1.0$, $\delta = 1.0$, as shown in figure 5, using an optimization scheme discussed in appendix A. The region where the tube totally collapses (when opposite walls of the tube come into contact) is of particular interest. This is indicated in figure 5 by a shaded region. From the figure, we are able to obtain the following approximate relationships for $A_c$ and $p_c$ when the tube fully collapses, which we denote $A_c^*$ and $p_c^*$:

$$\frac{A_c^*}{\pi a^2}\frac{l}{l_0} = 0.27 \pm 0.07, \tag{4.1a}$$

$$\frac{p_c^*}{E/(1-\nu^2)}\left(\frac{l}{l_0}\right)^{-2}\left(\frac{h}{a}\right)^{-1}\left(\frac{l_0}{a}\right) = -0.12 \pm 0.02. \tag{4.1b}$$

These equations allow us to find the collapse pressure and area for a flexible tube of arbitrary geometry, including the airways in the lung, and figure 5 gives a tube law directly applicable to the airways.

# 5. Fluid law and equilibrium

We now ask the question, if the flexible tube is held in a deformed state such that $A_c$ is known, and the suction $p_d$ is fixed, what will $p_c$ be? We call the function that describes this relationship the fluid law. Detailed calculation of the fluid law is shown in appendix B. Using this, we obtain fluid laws of the type shown in figure 6, which can be intuitively understood as follows.

If we assume that the Bernoulli equation holds between the points where $p_1$ and $p_2$ are measured, with cross-sectional areas $A_1$ and $A_2$, respectively, then

$$p_1 + \frac{1}{2}\rho\left(\frac{Q}{A_1}\right)^2 = p_2 + \frac{1}{2}\rho\left(\frac{Q}{A_2}\right)^2.$$

Because $A_1 = A_2$, $p_1$ and $p_2$ will be equal regardless of $A_c$. It can also be shown that the flowrate through the rig $Q$ will be independent of $A_c$, for a given value of $p_d$. In this case, $p_c$ will fall as $A_c$ is decreased

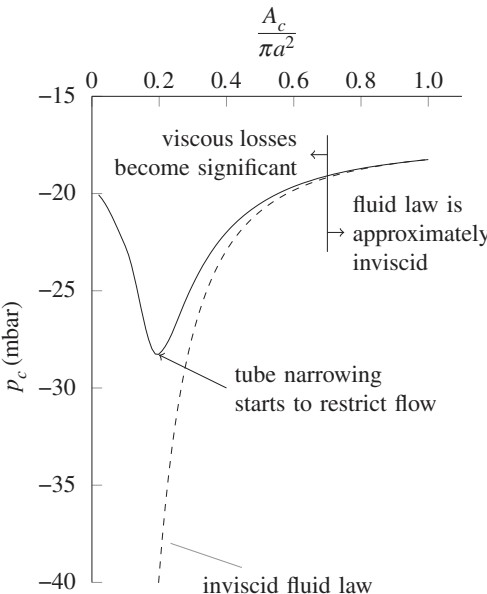

**Figure 6.** An example fluid law, calculated according to the method in appendix B. The solid curve represents the pressure $p_c$ as a function of $A_c$ for a fixed value of $p_d$. The dashed curve is a 'lossless' fluid law calculated assuming that the Bernoulli equation is applicable from the inlet to the outlet of the flexible tube. This fluid law is based on a tube for which $l = 23$ mm, $a = 3$ mm and $h = 0.35$ mm, and assumes that the air density is 1.2 kg m$^{-3}$.

owing to Bernoulli:

$$p_c = p_1 + \frac{1}{2}\rho\left(\frac{Q}{A_1}\right)^2 - \frac{1}{2}\rho\left(\frac{Q}{A_c}\right)^2,$$

and will tend to negative infinity as $A_c$ tends to zero. In reality, there is some degree of viscous and mixing loss in the tube, but when $A_c$ is close to $A_2$, i.e. the tube is almost fully open, this loss is small. Therefore, as $A_c$ is lowered from $A_2$, to begin with $p_c$ falls according to Bernoulli. As $A_c$ is lowered further, there comes a point when the flow separates downstream of the narrowest section of the tube, and viscous losses become much more significant. At this point, the flexible tube offers a significant obstruction to the flow, with significant pressure loss from $p_1$ to $p_2$. As $A_c$ is further reduced, this becomes large enough to cause the flowrate to drop, and $p_c$ starts to increase, as seen in figure 6. This explains the fluid law for a single value of $p_d$. Fluid laws for different values of $p_d$ are shown in figure 7, along with a tube law based on figure 5. Points of intersection between the tube and fluid laws yield equilibrium points. For a given suction, $p_d$, there is one equilibrium point. It can be seen that as $p_d$ decreases (suction increases), the equilibrium point moves in a direction that reduces $A_c$, the area at the narrowest point of the tube.

# 6. Static stability

To understand the static stability of an equilibrium solution, we start with $A_c = A_{c_0}$, given $p_d$, and then consider a small perturbation by lowering $A_c$. In general, this will change the value of $p_c$ according to the fluid law. If the new value of $p_c$ that is achieved is lower than the value required for equilibrium of the tube according to the tube law, then the tube will collapse further, leading to instability, but if it is higher, then the tube will return to the original equilibrium leading to stability. Hence, the system will be statically unstable if

$$\left.\frac{\partial f_f}{\partial A_c}\right|_{A_c = A_{c_0}} > \left.\frac{\partial f_t}{\partial A_c}\right|_{A_c = A_{c_0}}. \tag{6.1}$$

From figure 7, we can see that none of the equilibrium points are statically unstable. Hence, the onset of oscillations must depend on the dynamical behaviour of the system.

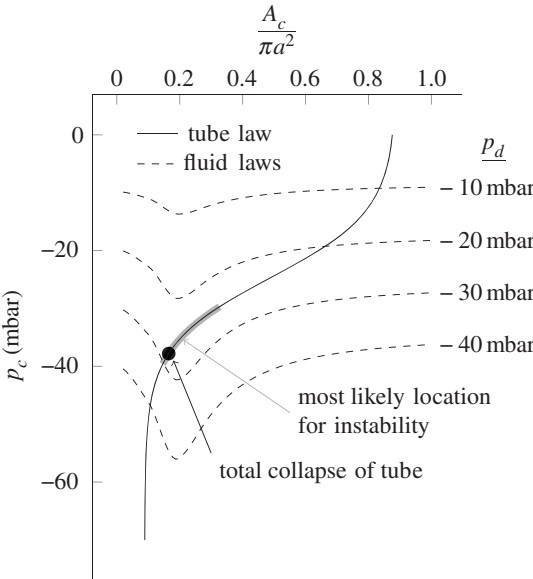

**Figure 7.** Tube law and fluid laws. The tube law is based on figure 5, and the fluid law is calculated according to the method presented in appendix B. Four fluid laws are shown for four different values of $p_d$ (dashed curves). Equilibrium solutions of the system as a whole correspond to intersections between the tube law and fluid laws.

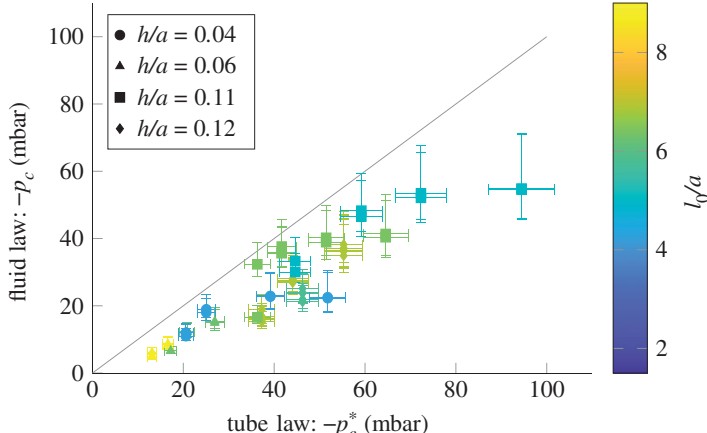

**Figure 8.** Predicting the choke point pressure at the onset of oscillations. Each point corresponds to one of the tubes shown in figure 1b. The pressure inside the tube at the onset of oscillations is plotted against the pressure required for total collapse of the tube according to the tube law. The colour of the marks indicates the unstrained tube length to radius ratio. Note that $-p_c$ is not measured directly but is derived from experimental results.

## 7. Mechanism for onset of oscillations

Our first observation from high-speed video is that the tube is nearly collapsed at onset. If onset happens close to total collapse of the tube, the $p_c$ measured from experiments at onset will be close to $p_c^*$ from equation (4.1b). Figure 8 shows $-p_c$ plotted against $-p_c^*$, and we can see that $-p_c$ is generally slightly below $-p_c^*$. Hence the onset of oscillations happens just before total collapse, in the region indicated in figure 7. It is significant that this shaded region also corresponds to the cross-sectional areas for which the flow within the tube is separating for the first time.

Without flow through the tube onset will not occur. We expect the dynamic pressure of the fluid to be comparable to $|p_c^*|$, otherwise the flow of fluid would not be capable of significantly influencing the shape of the tube. We can estimate the dynamic pressure in the narrowest part of the tube, $1/2\rho v_c^2$ assuming that $A_c = A_c^*$ (see appendix B). Figure 9 shows $1/2\rho v_c^2/|p_c^*|$ for various tubes. We can see that our results collapse onto a narrow range of values and combined with figure 8 we can predict the conditions for onset. The transmural pressure must be such that the tube is close to total collapse, and the flowrate

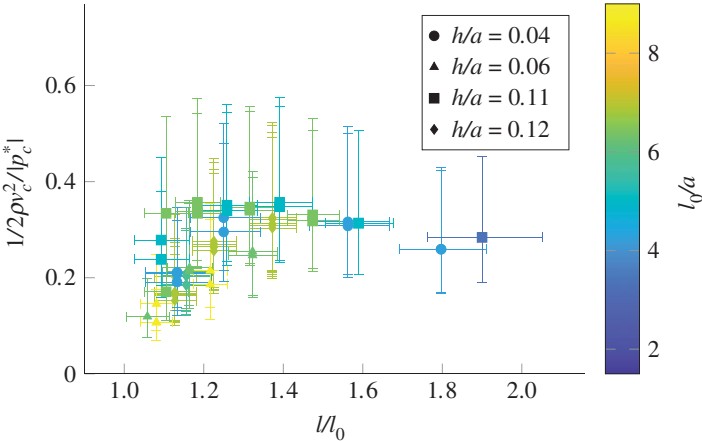

**Figure 9.** Predicting the flowrate at the onset of oscillations. Each plot point corresponds to one of the tubes shown in figure 1b. The horizontal axis of the plot is the axial strain of the tube, and the colour of the mark indicates the unstrained tube length to radius ratio. The vertical axis shows the ratio of the estimated dynamic pressure at the narrowest point of the tube $1/2\rho v_c^2$ to the collapse pressure of the tube $|p_c^*|$.

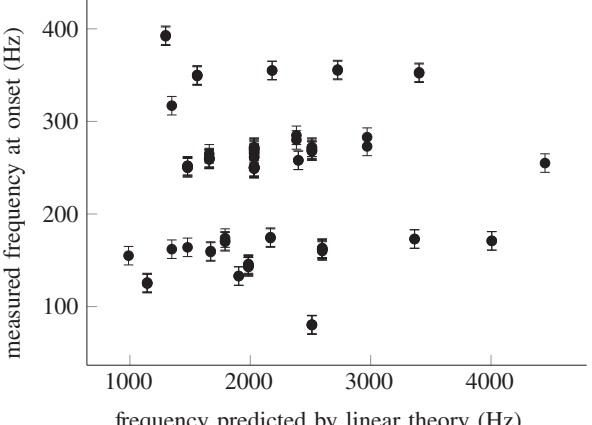

**Figure 10.** The failure of linear theory to predict the frequency of self-excited oscillations of flexible tubes. The first mode of oscillation of a cylinder, assuming an azimuthal mode number of 2, can be calculated given the geometric parameters $l$, $l_0$, $h$, $a$, and material properties $E = 1$ MPa, $v = 0.5$, $\rho_s = 1000$ kg m$^{-3}$, using linearized shell theory [55]. This prediction is compared with the frequency of self-excited oscillations measured at onset, showing no correlation. Error bars represent the variability of the measured frequency at onset.

of fluid through the tube must be such that the dynamic pressure at the narrowest point of the tube is comparable to the collapse pressure.

The final significant observation based on the video is that longitudinal waves are seen to be in sync with the opening and closing of the tube.

We see that in the time it takes for the longitudinal waves to complete $n$ cycles (travel up and down the tube $n$ times), the tube opens and closes once. $n$ is either 1 or 2 for the tubes investigated here. If $v$ is the speed of the downstream travelling wave and $u$ is the speed of the upstream travelling longitudinal wave, and $f$ is the frequency of oscillations, then mathematically we can represent our observation as $1/f = n(l/v + l/u)$. Therefore, the modified Strouhal number $f(l/v + l/u)$ must be equal to $1/n$ for $n = 1, 2, \ldots$.

In order to gain further insight and to be able to predict the frequency of oscillations at onset, we need a model to predict $v$ and $u$.

One hypothesis might be that these longitudinal waves are simply elastic waves propagating along the tube, and have nothing to do with the fluid. To test this hypothesis, we can calculate the resonant modes of a cylindrical tube held in tension, which correspond to elastic standing waves, with the geometries shown in figure 1b, and compare the frequencies of these modes to the frequencies observed at onset. The method used to calculate these modes is given in [55], and the results are shown in figure 10. These frequencies are neither in the correct range, nor do they correlate well with

royalsocietypublishing.org/journal/rsos　R. Soc. Open Sci. **8**: 201951

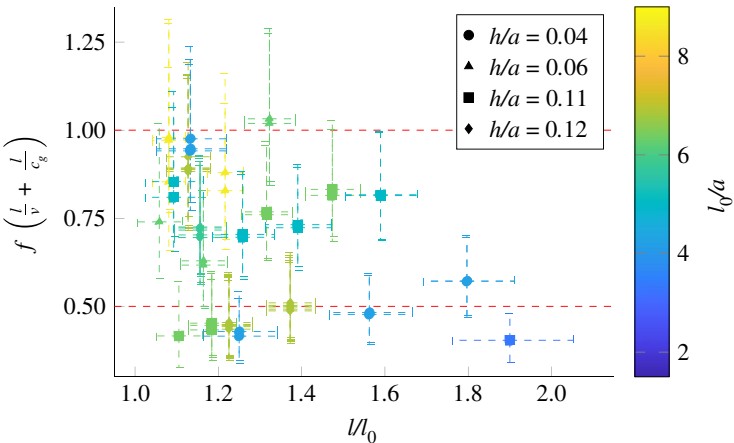

**Figure 11.** Predicting the frequency at the onset of oscillations. Each plot point corresponds to one of the tubes shown in figure 1b. The horizontal axis of the plot is the axial strain of the tube, and the colour of the mark indicates the unstrained tube length to radius ratio. The frequency of oscillations at onset ($f$) is compared to the frequency of longitudinal waves travelling at $v$ downstream and $c_g$ upstream. $v$ is estimated using the flowrate $Q$ and $c_g$ is that of an axisymmetric wave in a cylinder. Horizontal dashed lines indicate the criteria $f(l/v + l/c_g) = 1/n$.

the observed frequencies. This suggests that the waves we see travelling up and down the tube cannot be purely elastic.

We also know that the onset of oscillation is associated with the separation of the flow within the tube which would induce large scale turbulent structures to convect with the flow. Therefore, we hypothesize that the downstream travelling wave is driven by a flow feature such as a vorticity wave, while the upstream travelling wave is an elastic one. We can get an order of magnitude estimate of the speed of the downstream travelling wave using the bulk velocity $v = Q/\pi a^2$, and the speed of the upstream travelling wave using the group velocity of a longitudinal small amplitude wave travelling along a cylinder $u = c_g$. $c_g$ is found by modelling the tube as a thin walled cylindrical shell undergoing small (i.e. linear) deformations. The dynamical equations are derived based on [55], and we assume that the deformations are harmonic in time and space. Considering the second azimuthal mode, which best fits the shape of the deformations observed, the dynamical equations give a dispersion relation, from which the group velocity can be found. In figure 11, we have plotted $f(l/v + l/c_g)$ calculated using the measured frequency at onset. It shows that $f(l/v + l/c_g)$ has the correct approximate value. Therefore, we conclude that the downstream and upstream travelling longitudinal waves can be modelled by convected flow structures and elastic waves, respectively.

Based on our observations, we propose that the onset of oscillations in the tube is a result of an aeroelastic flutter. There is resonance between two modes of oscillations: a longitudinal mode (waves travelling up and down the tube) and a transverse mode (opening and closing of the tube). For flutter instability, the frequency associated with these two modes must match. To understand how this is possible, we return to figure 7 and note that onset occurs in a region of the tube law where the gradients of the fluid and tube law vary significantly. The difference between these gradients is effectively a stiffness for transverse oscillations, and so in this region the resonant frequency of transverse oscillations will vary significantly. Therefore, oscillations will start when the resonant frequency of transverse oscillations matches that of the longitudinal waves. The other place on the tube law that this could happen is when $|p_d|$ is small such that the tube just starts to close, but here the flow inside the tube will not have separated, and so there is no way for the flow to drive the longitudinal wave, and hence the oscillations cannot be self-excited.

# 8. Applying the mechanism to the lung

We have seen that, in order for self-excited oscillations to happen, the transmural pressure across the tube wall must be large enough to bring the tube to near total collapse (figure 8), and the flow through the tube must result in a dynamic pressure comparable to the collapse pressure of the tube (figure 9).

To apply these observations to the lung, we need a model of airflow through the lung. This is a topic of ongoing research, but based on simple models, we can draw some interesting conclusions. For the

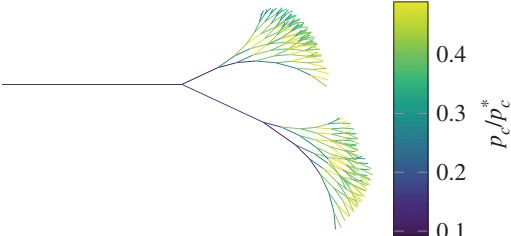

**Figure 12.** The ratio of the transmural pressure across and airway to the pressure required to collapse the airway. The trachea is to the left, with the airways branching into bronchi and bronchioles to the right.

geometry of the airways, we use the Horsefield model for the human lung [9,10,56] which is described in more detail in appendix C. During normal relaxed breathing the pressure drop from the alveoli and alveolar ducts to the mouth is up to $20\,cmH_2O \approx 2000\,Pa$ [57]. Maximum expiratory pressure can be significantly larger, varying considerably depending on the size of the person from $39\,cmH_2O$ to $100\,cmH_2O$ [58,59]. The maximum expiratory pressure is obtained by breathing out as hard as possible. In asthmatic patients, the pressure drop for normal breathing is also raised, [60], as can be inferred by the presence of intrinsic positive end expiratory pressure.

For any of these situations, we can estate the maximum transmural pressure seen by any individual airway in the lung, which corresponds to $-p_c$. We can compare this to $-p_c^*$ to see how close any given airway is to collapse. In figure 12, we plot $p_c/p_c^*$ throughout the first eight generations of the airways, assuming that $-p_c = 20\,cmH_2O$, which is the maximum possible value of $-p_c$ for normal relaxed breathing. In reality, $-p_c$ will be smaller than $20\,cmH_2O$ because of pressure drops in the tubing, so we use $-p_c = 20\,cmH_2O$ as an upper bound. From this, it is clear that even using an upper bound for $-p_c$ the transmural pressure is too small for collapse of the tube during normal breathing, which essentially rules out wheezing during normal breathing, which is what is observed in reality [2]. However, by forcing expiration $-p_c$ can be more than doubled, and figure 12 indicates that this would be enough to collapse several of the airways, opening the possibility of wheezing. This is what is observed and is usually called 'forced expiratory wheeze'. The increase in $-p_c$ associated with asthma could partially explain why it is so often associated with wheezing [2,60], however, asthma also introduces obstruction, which we will discuss later in this section.

To calculate $p_c^*$ in figure 12 we used equation (4.1b), the geometry of each tube of the airway found from the Horsefield model (appendix C), and an estimate of the axial strain in the airways from [10]. The elastic properties of lung tissue are estimated by [44–47]. We take the tissue to be incompressible with Young's modulus of 1 MPa.

We know that during normal expiration the flowrate of air out of the lungs reaches between $6.5\,l\,s^{-1}$ and $11.5\,l\,s^{-1}$ [61]. Given a particular distance into the lung ($x$), we can find every distinct path into the lung of that length. The tubes at the end of these paths will have a cross-sectional area, and we can sum all of these areas to obtain a cumulative cross-sectional area as a function of distance into the lung, which we denote $A(x)$. We denote the velocity at the top of the trachea $v_i$, and the area at the top of the trachea $A_i$. A simple model for the velocity of the fluid at a distance $x$ into the lung is then,

$$v(x) = \frac{A_i}{A(x)} v_i. \tag{8.1}$$

Using this we can estimate $1/2\rho v_c^2/|p_c^*|$ throughout the lung. Note that we have assumed that the tubes are all uncollapsed. $1/2\rho v_c^2/|p_c^*|$ is plotted in figure 13, showing values much smaller than those consistent with self-excited oscillation in figure 9. This only applies for a fully open tube, which we expect during normal relaxed expiration. If expiration is forced the tube can collapse, reducing the area, and hence increasing $1/2\rho v_c^2/|p_c^*|$. If the area reduces by a tenth, which is roughly consistent with tube collapse, we expect $1/2\rho v_c^2/|p_c^*|$ to increase by a factor of around one hundred, which would put it in a regime where self-excited oscillation is possible. Once again, this is consistent with the observation that wheezing is only seen in forced expiration.

Comparing figures 8 and 12, and figures 9 and 13, we see that under normal breathing conditions wheezing is unlikely to happen. In order for wheezing to happen we need both the transmural pressure and the flow velocity to be large enough, and the trade off between these two requirements indicates that wheezing is most likely to occur in the first few branches of the bronchioles, which is where it is generally believed that most wheezing originates [2].

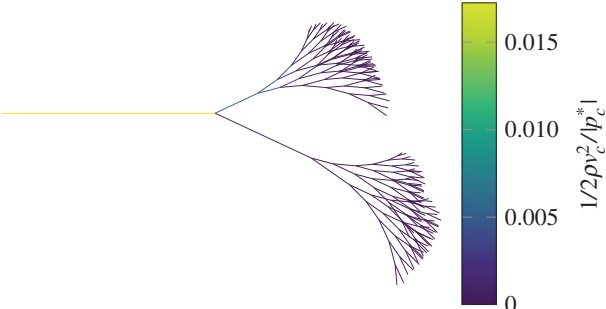

**Figure 13.** The ratio of the dynamic pressure of the fluid within the airway to the pressure required to collapse the airway.

It has been observed many times that obstruction within the airways is associated with wheezing [5]. We can explain this in two ways. Obstruction will result in greater pressure loss of air along a given tube, lowering the internal pressure in tubes downstream (towards the mouth) of the obstruction, and so pushing the tube closer to collapse, and oscillation. The obstruction itself will also partially close the tube, increasing the fluid velocity at that location, making total collapse require a smaller transmural pressure difference. This is supported by the qualitative observation made during experiments that if a tube was partially collapsed, it could be made to oscillate by gently pinching it closed.

There is another conclusion that we may draw from our analysis relating to the role played by axial tension in the airways. As the breathing cycle progresses, the axial tension that the airways are held in varies from a maximum at the beginning of the exhale to a minimum at the end of it. We know from the tube law that increasing the axial tension increases the transmural pressure needed to collapse the tube (figure 5). Hence, we would expect that wheezing would become easier towards the end of the exhale when the axial tension of the airways is lower. This is what we see in forced expiratory wheezes, which can be produced by most healthy people by breathing our hard through the mouth. These wheezes tend to start around halfway through the exhale, and continue to the end of the breath.

Our analysis of wheezing in the paper has focussed on the onset of the oscillations and their characterization. How the vibrations past the onset produce the wheezing that can be heard on the chest, for example by a stethoscope, is not an objective of this paper. However, based on our work, we are able to comment on the nature of the acoustic sources of wheezing sounds. The oscillating tube produces a varying mass flux through the airway, which will act as a sound source of a monopole type, and the moving walls themselves will act as a sound source, though because of how the walls move, possibly a less efficient one of a dipole or quadrupole type ([62], §4). It is also necessary to understand how the sound 'gets out' of the chest because doctors use sounds measured at the chest to make diagnoses. Understanding of these phenomena could help make diagnosis more specific and be an interesting avenue for further research.

## 9. Conclusion

We have used self-excited oscillations of stretched elastic tubes driven by an air flow (a modified Starling resistor) as a model for wheezing in lungs. We have performed experiments with a wide range of flexible tubes with properties directly applicable to the lungs—short rubber tubes of various thicknesses and lengths held in various degrees of axial tension. Through a novel use of multiple-camera stereoscopy, we were able to produce the first tube law for such tubes. Using dimensional analysis and the extensive set of data collected in the study, we were able to collapse the data to derive a generalized tube law that allows us to predict the cross-sectional area at the narrowest point in the tube as a function of transmural pressure for any tube given its material properties, geometry and the amount of axial tension.

Our results show that, for onset of oscillations there are two necessary conditions: the transmural pressure should be such that the tube is nearly collapsed; and the dynamic pressure through the narrowest cross-section of the tube should be similar in magnitude to the transmural pressure required to collapse the tube. Hence, a knowledge of the tube law is essential in capturing the conditions necessary for onset.

We have proposed a model for the self-excited oscillations based on aeroelastic flutter in which two modes of vibration - longitudinal and transverse - couple to give resonance. The model is

phenomenological and further work is necessary to fully understand the physical mechanism for the oscillations. To make further progress, a better understanding of the flow field inside the oscillating tubes is necessary. Our experiments could be supplemented with a numerical study to help guide such an effort.

Our work has interesting implications for wheezing in the lungs. For instance, based on the frequency of wheeze, it might be possible to pinpoint the generation of bronchiole responsible for the wheeze. It may also be possible to predict the material properties of the bronchiole. Therefore, an understanding of the tube laws and the physical mechanism of wheezing would lead to better diagnosis and treatment of lung diseases.

Data accessibility. Data from self-excited oscillations of flexible tubes is available at [63,64], and data from experiments measuring the tube law is available at [65]. High-speed video recordings of self-excited oscillations are available at [50]. Some videos are also available on our group website acoustics.eng.cam.ac.uk.
Authors' contributions. The work was completed by A.G., supervised by A.A. and J.L.
Competing interests. We declare we have no competing interests.
Funding. This work was supported the EPSRC, the IMechE Post-graduate Research Scholarship, Engineering for Clinical Practice (http://divf.eng.cam.ac.uk/ecp/Main/EcpResearch), and the Cambridge Philosophical Society.
Acknowledgements. Experimental work was made possible by the senior laboratory technicians John Hazlewood and Dave Martin. We thank Michael Sutcliffe and Alex Casabuena-Rodriguez allowed for their help using an Instron machine. We are grateful to Will Graham and Stephan Bansmer for their comments on an earlier draft of this manuscript.

# Appendix A. Calculating the tube law

Figure 14 illustrates the rig used to find how the cross-sectional area at the narrowest point of a flexible tube ($A_c$) changes as the pressure inside ($p_c$) is lowered relative to the outside pressure. The pressure within the tube is gradually pumped down using a syringe and monitored with a Digitron 2028P manometer. At a set of fixed internal pressures, multiple images are taken of the tube using a calibrated camera system, which are then used to reconstruct the shape of the collapsed tube. This allows us to calculate the cross-sectional area of the tube.

The camera system used is illustrated in figure 14. It allows a camera to be positioned in one of seven locations circling the tube, separated by 45°. In order to calibrate this camera system, we must know the internal parameters of the camera being used, and the rotations and translations from the first camera position to the other six. The camera used is an Olympus SP-560 UZ, with an f-number of 7.1, a shutter speed of $1/100$ s, a 50 mm zoom, an ISO of 50, and is manually focused at 10 cm.

Multiple images were taken of chequerboards from all seven camera positions. This process produced a set of $n \approx 2000$ distinct points in the world (three-dimensional space). For each point and camera, there is an associated position in the image if the point is visible.

In order to obtain physical coordinates (in millimetres) from pixel coordinates in each camera, we need to find the intrinsic properties of the camera [66]. This was done using all of the chequerboard images and the single-camera calibration program in the Matlab computer vision system toolbox [67].

To triangulate using our camera system, we also need to know the relative rotations and translations between each camera position. This is known as externally calibrating the system. Accurate rotations and translations are found by minimizing the error function proposed by Lasenby & Stevenson [68]. This error function is the sum of squared distances between the world points and their closest points on the camera rays projecting out from the observed image points. An in-house code was used to perform the external calibration.

With the cameras calibrated, the cross-sectional area of the flexible tubes could be measured as a function of the $p_c$. Some example cross-sectional reconstructions are shown in figure 15. Uncertainty in the cross sectional shape can be seen in the spread of points in this figure, and this is the basis of the horizontal error bars in figure 5. For more information on the reconstruction of the tube cross sections, see [48]. As was shown in the main article, we are interested in how the cross-sectional area as a function of $p_c$ changes as the geometric parameters $l_0/l$, $h/a$ and $l_0/a$ are varied. The tube law was measured for 11 different tubes, whose geometric parameters are given in table 1.

For each tube, we obtain a set of $A_c$ values with a corresponding set of $p_c$ values. Each pair of sets represent a single plot of $p_c$ against $A_c$, and for each of these plots, there is an associated value of $l/l_0$, $h/a$ and $l_0/a$. We wish to find the values of $\alpha$, $\beta$, $\gamma$ and $\delta$ such that when we plot

$$\frac{A_c}{\pi a^2} \left(\frac{l}{l_0}\right)^\alpha, \tag{A 1}$$

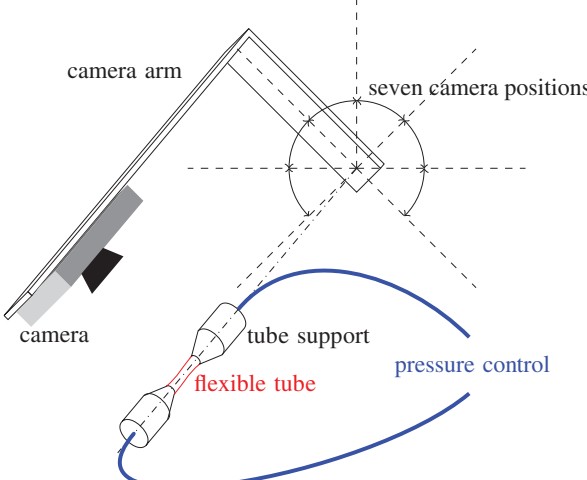

**Figure 14.** Experimental rig to investigate tube collapse. Suction is used to lower the pressure inside the flexible tube ($p_c$) relative to the environment ($p_e$) in a controlled way, and the cross-sectional area is measured by taking images from the seven camera positions shown.

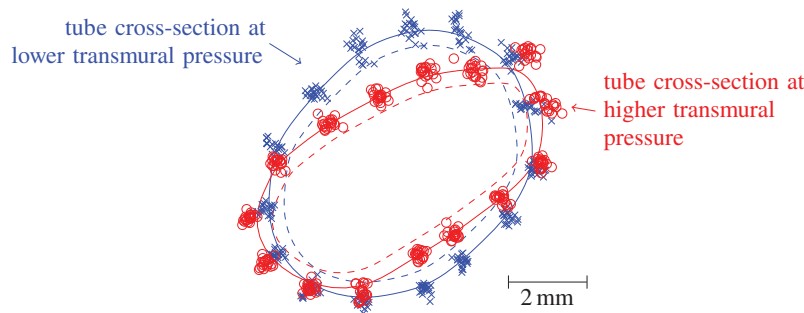

**Figure 15.** Example cross-sections of tubes reconstructed using the camera rig in figure 14. The solid lines are fitted to the collections of points, which are estimates of the three-dimensional positions of dots on the tube. The scattering of the dots represents the uncertainty in their three-dimensional position. The dashed lines are our estimate of the inner surface of the tube.

**Table 1.** Geometric parameters of tubes used to measure the tube law. ($l_0$ is the unstrained length of the tube, $l > l_0$ is the length the tube is stretched to in the experiment, $h$ is the input tube wall thickness and $a$ is the unstrained inner radius of the tube.)

| tube number | $l/l_0$ | $h/a$ | $l_0/a$ |
|---|---|---|---|
| 1 | 1.01 | 0.12 | 6.03 |
| 2 | 1.11 | 0.12 | 6.03 |
| 3 | 1.21 | 0.12 | 6.03 |
| 4 | 1.28 | 0.12 | 6.03 |
| 5 | 1.09 | 0.06 | 6.59 |
| 6 | 1.17 | 0.06 | 6.59 |
| 7 | 1.26 | 0.06 | 6.59 |
| 8 | 1.10 | 0.12 | 4.81 |
| 9 | 1.23 | 0.12 | 4.81 |
| 10 | 1.35 | 0.12 | 4.81 |
| 11 | 1.48 | 0.12 | 4.81 |

against,

$$\frac{p_c}{E/(1-\nu^2)}\left(\frac{l}{l_0}\right)^{\beta}\left(\frac{h}{a}\right)^{\gamma}\left(\frac{l_0}{a}\right)^{\delta}, \tag{A 2}$$

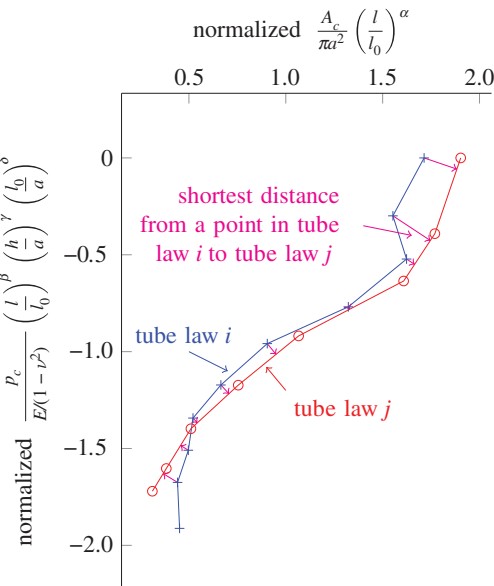

**Figure 16.** The difference between two tube laws. To find the difference between tube law $i$ and $j$, we find the shortest distance from each point in tube law $i$ to tube law $j$. The average of these distances provides a measure of how different the tube laws are.

the tube laws collapse onto a single curve. We have chosen to look for this functional form based on the observation that the tube law always has the same basic shape; it is either stretched or compressed along the pressure and area axes depending on the values of $l/l_0$, $h/a$ and $l_0/a$. Specifically, increasing $l/l_0$ results in compression along the area axis and stretching along the pressure axis, increasing $h/a$ results in stretching along the pressure axis, and increasing $l_0/a$ results in compression along the pressure axis. These observations mean that we expect to have $\alpha > 0$, $\beta < 0$, $\gamma < 0$ and $\delta > 0$.

The values of $\alpha$, $\beta$, $\gamma$ and $\delta$ are those that minimize the distance between the curves we obtain when we plot expression (A 1) against expression (A 2). We consider two tube laws, $i$ and $j$, and assume that we have an estimate of the values $\alpha$, $\beta$, $\gamma$ and $\delta$. Then we plot expression (A 1) against expression (A 2) and normalize the axes so that the mean value in both directions is 1, and obtain a plot such as the one shown in figure 16. Next, we find the shortest distance from each point in tube law $i$ to tube law $j$. When considering more than two tube laws, we find the shortest distances for all possible pairs, and then find the overall average minimum distance. We run a nonlinear optimization to minimize this average shortest distance using the `fminunc` function in Matlab [67]. The results of this optimization were $\alpha = 1$, $\beta = -2$, $\gamma = -1$, $\delta = 1$, which results in the plot shown in figure 5. Note that the integer values are not special, they are simply the values that fell out of the optimization to a precision of one decimal place. A higher precision than this was found to be redundant given the spread of the experimental data in figure 5. This is the first generalized tube law valid for relatively short tubes held in axial tension.

# Appendix B. Calculating the fluid law

To understand the equilibrium of the fluid, we break down the overall pressure drop through the rig, $-p_d$, into four parts:

$$-p_d = -p_1 + (p_1 - p_c) + (p_c - p_2) + (p_2 - p_d). \tag{B 1}$$

Note that the pressures are measured relative to the external pressure $p_e$. Either side of the flexible tube are a set of components (settling chambers, tubes, flow metres, contractions, expansions) for which the relationship between the flowrate through them and the pressure drop across them is predictable [69]. Upstream of the flexible tube we characterize the overall relationship between $-p_1$ and $Q$ by the function $-p_1 = f_u(Q)$. This includes pressure drops owing to the flow metre (2), upstream settling chamber (3), clean flow inlet (4), upstream clean flow tube (5) and contraction (6), where the numbers in parentheses refer to locations in figure 2. Similarly, downstream of the flexible tube, the overall relationship between $p_2 - p_d$ and $Q$ is characterized by $p_2 - p_d = f_d(Q)$, which includes pressure drops owing to the expansion (6'), downstream clean flow tube (5'), and the sudden expansion into the

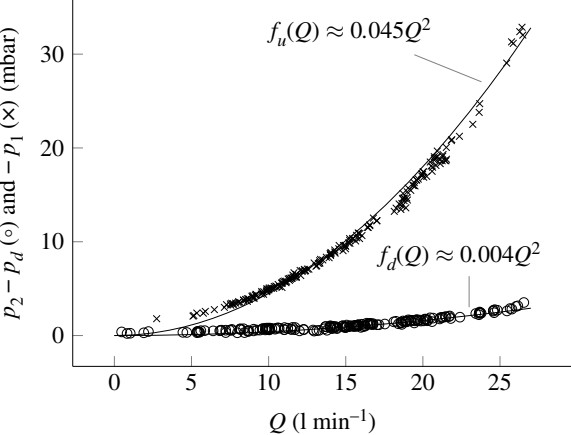

**Figure 17.** Pressure-flow relationships for the experimental rig in figure 2. The relationship between $-p_1$, $p_2 - p_d$ and $Q$ is shown for situations where flow through the rig is steady. For the plots shown here, the length of the upstream clean flow tube was 94.8 cm, and the length of the downstream clean flow tube was 44.0 cm. The plotted points are obtained from multiple experimental runs before the onset of oscillations.

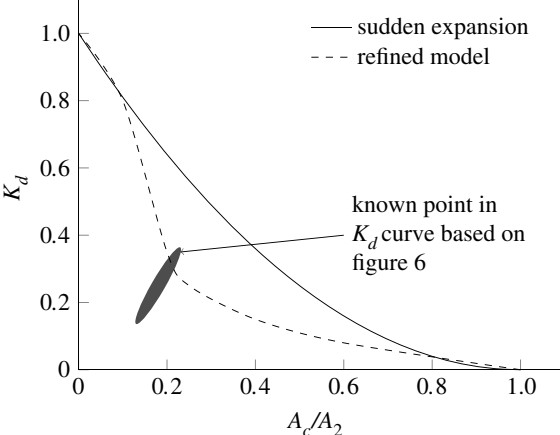

**Figure 18.** Dependence of the loss coefficient $K_d$ on the area ratio $A_c/A_2$. The loss coefficient $K_d$ that determines the pressure change $p_c - p_2$ depends on the area ratio between the narrowest point of the tube and the point where $p_2$ is measured. We plot the dependence for the sudden expansion model along with a refined model based on Miller [69] and results from figure 3. Note that, in our rig, $A_2 = \pi a^2$.

downstream settling chamber (3′). These functions could be found by considering data in, for example, the work of Miller [69], but it is simpler for us to find these functions empirically directly from our experimental data recorded before the tube starts to oscillate, as shown in figure 17.

To completely characterize the rig we need to know how $p_1 - p_c$ and $p_c - p_2$ depend on $Q$, $A_c$, and other geometric parameters of the flexible tube. Following the methodology of [69], these relationships can be written as:

$$p_1 - p_c = f_1(Q, A_c) = \frac{1}{2}\rho Q^2((1 + K_c)A_c^{-2} - A_1^{-2})$$

and

$$p_c - p_2 = f_2(Q, A_c) = \rho\frac{Q^2}{A_2}(A_2^{-1} - (K_d - 1)A_c^{-1}).$$

These equations are Bernoulli with loss coefficients $K_c$ and $K_d$, which in general are a function of $A_c$ and the flexible tube geometry. $K_c = 0$ or $K_d = 0$ imply that the Bernoulli equation holds. In reality, both of the coefficients will be greater than zero. Miller [69] gives a wide range of information on loss coefficients. For the contraction from $A_1$ to $A_c$, $K_c$ will be between 0.01 and 0.1, and its effects on the overall results

of this article are not significant. Hence we will assume $K_c = 0.05$. If we model the expansion, from the narrowest point of the tube ($A_c$) to the point where $p_2$ is measured, as a sudden expansion in space (the flow separates as a jet past the narrowest point and reattaches by $A_2$), then it can be shown by an application of conservation of mass and momentum to an appropriate control volume that $K_d = 1 - 2(A_c/A_2) + (A_c/A_2)^2$. $K_d$ as a function of $A_c/A_2$ for a sudden expansion is shown in figure 18. Bernoulli and sudden expansion are the two extremes. Bernoulli implies no loss and sudden expansion implies maximum loss because of mixing. The flow conditions in the segment between $A_c$ and $A_2$ are somewhere in between these extremes. Empirical information for the loss coefficient $K_d$ is available from Miller [69] for area ratios $A_c/A_2 > 0.25$, but the data does not cover the range of area coefficients needed in the present study ($0.1 < A_c/A_2 < 1$). Therefore, we must supplement the data by our own observations. We can find an approximate point on the curve of $K_d$ against $A_c/A_2$ by noting that when oscillations start, the tube is almost fully collapsed, and so $A_c/A_2 \approx 0.2$. We know $p_1 - p_c = f_1(Q, A_c)$, and we have measured $p_1$ and $p_2$ at onset in multiple experiments (figure 3), so we can find a single value of $K_d$ corresponding to $A_c/A_2 \approx 0.2$. This is shown in figure 18 as a shaded ellipse. The dashed curve in figure 18 combines this empirical point with the data from Miller [69]. The dashed curve is a piecewise cubic interpolation through the points $(A_c/A_2, K_d) = (0, 1), (0.1, 0.8), (0.2, 0.35), (0.25, 0.25), (0.4, 0.15),$ $(0.6, 0.08), (1, 0)$. With this we have completed the definition of $p_c - p_2 = f_2(Q, A_c)$.

Now equation (B1) can be written as:

$$-p_d = f_u(Q) + f_1(Q, A_c) + f_2(Q, A_c) + f_d(Q). \tag{B2}$$

Given a value for $p_d$ and $A_c$, we can solve this equation to find $Q = Q(p_d, A_c)$. Then using this value of $Q$ we can find $p_c = -f_u(Q) - f_1(Q, A_c)$, and the desired fluid law, $p_c = f_f(A_c, p_d)$. Examples of this fluid law are shown in figure 6. The fluid law has two main regions. When $A_c/A_2 > 0.3$ there is little separation, and $K_d$ is small. This changes abruptly when the curve in figure 18 turns upwards at $A_c/A_2 \lesssim 0.25$. This additional loss means that the obstruction to the flow posed by the flexible tube becomes significant, and the flowrate drops as $A_c$ is reduced, meaning that $p_c$ increases.

We are interested in the value of $p_c$ at the onset of oscillations, but this is not measured directly. $p_1$ and $Q$ are measured though, so $p_c$ can be found using

$$p_c = p_1 - f_1(Q, A_c).$$

Hence we need to know $A_c$ in order to find $p_c$ from experimental measurements. From high-speed video, we know that the tube is strongly collapsed at onset, so to estimate $p_c$ we take $A_c = A_c^*$ from the tube law (equation (4.1a)).

# Appendix C. The Horsefield model

The Horsefield model for the human lung divides the airways into 35 different categories. For each category the airway length ($l$), diameter ($d$) and wall thickness ($h$) are given, along with $\Delta$, which defines the degree of asymmetry in the airway network. The categories were numbered by Horsefield as shown in table 2, with $n = 35$ corresponding to the trachea. The model assumes that the end of each airway (of category $n$) splits into two smaller ones, whose categories are given by $n - 1$ and $n - 1 - \Delta$. The final airways are assumed to each split into two alveoli.

**Table 2.** The Horsefield model for the human lung.

|  | $n$ | $l$ (cm) | $d$ (cm) | $h$ (cm) | $\Delta$ |
|---|---|---|---|---|---|
| trachea | 35 | 12.5 | 2 | 0.4655 | 1 |
|  | 34 | 6.25 | 1.5 | 0.2169 | 2 |
|  | 33 | 2.75 | 1.375 | 0.1685 | 3 |
|  | 32 | 1.375 | 1 | 0.066 | 3 |
|  | 31 | 1.3125 | 0.9125 | 0.0511 | 3 |
|  | 30 | 1.4125 | 0.7375 | 0.0305 | 3 |

(Continued.)

| | n | l (cm) | d (cm) | h (cm) | Δ |
|---|---|---|---|---|---|
| | 29 | 1.4125 | 0.7375 | 0.0305 | 3 |
| | 28 | 1.2125 | 0.675 | 0.0256 | 3 |
| | 27 | 1.35 | 0.5375 | 0.0186 | 3 |
| | 26 | 1.075 | 0.4375 | 0.0158 | 3 |
| | 25 | 1.1875 | 0.4375 | 0.0158 | 3 |
| | 24 | 1.2375 | 0.3875 | 0.0147 | 3 |
| | 23 | 1 | 0.3625 | 0.0143 | 3 |
| | 22 | 1.15 | 0.25 | 0.014 | 3 |
| | 21 | 1.025 | 0.3375 | 0.0139 | 3 |
| | 20 | 1.0125 | 0.3125 | 0.0134 | 3 |
| | 19 | 0.9625 | 0.3 | 0.0131 | 3 |
| | 18 | 0.8 | 0.2725 | 0.0125 | 3 |
| | 17 | 0.7875 | 0.25 | 0.012 | 3 |
| | 16 | 0.6462 | 0.225 | 0.0114 | 3 |
| | 15 | 0.6 | 0.2 | 0.0106 | 3 |
| | 14 | 0.525 | 0.175 | 0.0061 | 3 |
| | 13 | 0.45 | 0.1375 | 0.0084 | 2 |
| | 12 | 0.3875 | 0.1189 | 0.0075 | 2 |
| | 11 | 0.3125 | 0.095 | 0.0063 | 1 |
| | 10 | 0.1375 | 0.0788 | 0.0052 | 0 |
| | 9 | 0.1638 | 0.0663 | 0.0045 | 0 |
| | 8 | 0.1313 | 0.06 | 0.004 | 0 |
| | 7 | 0.0938 | 0.0537 | 0.0036 | 0 |
| | 6 | 0.0737 | 0.1 | 0.0065 | 0 |
| | 5 | 0.06 | 0.1 | 0.0065 | 0 |
| | 4 | 0.06 | 0.1 | 0.0065 | 0 |
| | 3 | 0.06 | 0.1 | 0.0065 | 0 |
| | 2 | 0.06 | 0.1 | 0.0065 | 0 |
| final airway | 1 | 0.06 | 0.1 | 0.0065 | 0 |

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
