## [Peer Review File · Royal Society Open Science]

Review History

RSOS-201951.R0 (Original submission)

Review form: Reviewer 1

Is the manuscript scientifically sound in its present form?

Yes

Are the interpretations and conclusions justified by the results?

Yes

Is the language acceptable?

Yes

Do you have any ethical concerns with this paper?

No

Have you any concerns about statistical analyses in this paper?

No

Recommendation?

Accept with minor revision (please list in comments)

Comments to the Author(s)

The manuscript is directed towards a physics-based understanding of the onset of self-excited oscillations in flexible tubes with implications on wheezing, i.e. sounds used for diagnosis of pulmonary diseases. The manuscript is clearly written, of high quality. I consider that it can be accepted after some minor revisions as suggested below.

Q1: One must mention that the analysis of wheezing is almost exclusively focused on the on-set of oscillations and their characterization, without a detailed analysis of the mechanisms for sound production. Will the authors comment on this aspect?

Q2: Which has been the main reason for going beyond the typical values of the geometric parameters found in the upper airways, but in the same time not exploring the entire space for h/d (see also Fig. 1)?

Q3: Can the authors comment on the reason for which the expiration phase have been considered? Does not makes sense to focus on both inspiration & expiration?

Q4: The movies are showing that the wall dynamics are significant, with quite a variation of the tube's minimal cross-sectional shape. Can one comment on the fluid flow characteristics associated with such a scenario and the potential impact on wall dynamics? Moreover, unsteady pressure loads on surfaces may also generate sound.

Q5: How the very rigorous analysis on understanding of onset of oscillations on the Starling resistor (of different topologies and under different conditions) can be used for assessing the lungs and their function (as a series of Starling resistors interconnected). How the developed model for predicting self-excited oscillations in bronchi can be used in the clinical world?

Q6: What one can say about the acoustic sources generating wheezing?

Q7: How one can ensure the transfer of information and knowledge to the physicians so that one can make sure that the developed model (for predicting onset of self-excited oscillations with relevance to wheezing in the lungs) can be used in diagnosis?

Review form: Reviewer 2 (Guilherme Garcia)

Is the manuscript scientifically sound in its present form?

Yes

Are the interpretations and conclusions justified by the results?

Yes

Is the language acceptable?

Yes

Do you have any ethical concerns with this paper?

No

Have you any concerns about statistical analyses in this paper?

No

Recommendation?

Accept with minor revision (please list in comments)

Comments to the Author(s)**MAJOR COMMENTS**

This manuscript investigates the physical mechanisms associated with wheezing (generation of sounds in the lungs). The authors report sophisticated *in vitro* experiments and a mathematical model of the fluid dynamics in collapsible tubes. Based on these results, they propose a phenomenological model that describes self-excited oscillations as an “aeroelastic flutter in which two modes of vibration - longitudinal and transverse - couple to give resonance” (Page 16, Line 37). From my knowledge of the literature in this field, this manuscript is a major contribution on this topic in many levels, including the description of novel tube laws that account for longitudinal stretching, very carefully conducted *in vitro* experiments, and the introduction of new concepts such as that self-excited oscillations are a “resonance between two modes of oscillations: a longitudinal mode (waves travelling up and down the tube) and a transverse mode (opening and closing of the tube)” (Page 13, Line 42).

My only major comment is that Section 8 (Applying the Mechanism to the Lung) lacks important details. The authors need to revise this section to provide more details:

(1) The authors state: “For the geometry of the airways we refer to [18,41,50].” The authors should include a table with the length, diameter, and wall thickness of each generation in their lung model.

(2) What modulus of elasticity (Young’s modulus) was used in the lung model? I imagine that the modulus of elasticity plays a major role given that a lung made of “steel” would not collapse. How was the modulus of elasticity of lung structures estimated?

(3) To create Figure 12, I assume the authors used a 1-dimensional model to estimate the pressure-flow relationship in their lung model, but this is not described. How was the pressure field in the lung model computed?

(4) How was the pc^* computed for each lung generation?

MINOR COMMENTS

I have several suggestions to improve readability.

Page 3, Line 15 - This sentence is a bit strange. Isn’t air always “the fluid flowing in the airways”? I assume that references [5,6] investigated other gases (helium?). But I recommend revising this sentence for better clarity.

Page 3, Line 39 - Elliott and Dawson [1] is another example of experimental work that used air as the working fluid.

[1] - Elliott E.A. and Dawson S.V. (1977) Test of wave-speed theory of flow limitation in elastic tubes. *J. Appl. Physiol.* 43(3), 516-522.

Page 4, Line 34 - Please specify the source of the rubber tubes used in the experiments. Also, please explain if the modulus of elasticity ($E = 1$ MPa) and Poisson’s ratio ($\nu = 0.5$) were measured in your lab or if the mechanical properties of rubber come from the tube manufacturer.

Page 4, Lines 36-45 – As the authors stated, “it is vital for experimentalists to describe, quantitatively, all aspects of their experiment” (Page 3, Line 29). In this spirit, I recommend that the authors report as a supplementary material all the dimensions of their experimental setup, such as the volumes of the settling chambers (3 and 3' in Fig. 2), the diameters and lengths of the upstream (5) and downstream (5') tubes, the dimensions of the contraction (6) and expansion (6'), etc.

Page 4, Lines 36-45 – How was the rubber tube clamped to the contraction and expansion? (I assume there was a slight change in diameter as the rubber tube was clamped over the contraction and expansion.) This information may be useful to provide the boundary conditions in future computational work that may try to reproduce this experiment.

Page 6, Line 18 – The following sentence is unclear because Figure 3 does not display the pressure difference $p_1 - p_2$: “This shows that the onset of oscillation introduces a pressure loss between p_1 and p_2 and restricts the flowrate.” Perhaps the authors can add a line to Figure 3a showing the pressure difference $p_1 - p_2$.

Page 6, Line 37 – Please replace “affect” with “effect”.

Page 6, Line 51 – The authors state that “Videos from our study can be seen in the supplementary materials for this paper.” I could not find supplementary materials on the Royal Society Open Science website. I watched the video corresponding to Figure 4 whose link is Ref. [59], but it is unclear what are the “supplementary materials” the authors are referring to in this sentence.

Figure 3 – The legend of panel (a) in Figure 3 shows a dashed line for p_1 and a solid line for p_2 . However, the plot shows dashed lines for both p_1 and p_2 . I assume that p_2 is the lower border of the solid colored area, but this is not entirely clear given the disconnect with the legend.

Figure 4 – Please explain in the legend of Figure 4 that the white dots on the rubber tube are light dots to improve visualization.

Page 8, Line 27 – I believe the variable “a” is used here for the first time without explaining that “a” is the radius of the undeformed tube.

Page 8, Line 41 – It is unclear why the term $(l/10)$ is included in both the left and right sides of the equation used to define the axes of Figure 5. Wouldn't it be easier to use $A_c/(\pi \cdot a^2)$ as the X-axis in Figure 5?

Figure 6 – Please provide in the legend of Figure 6 the tube geometry for which this fluid law refers to.

Page 11, Line 34 – Please replace the subsection title “Mechanism for Onset” with “Mechanism for Onset of Oscillations”. Likewise, please replace “onset” with “onset of oscillations” in line 35.

Figure 8 – Please replace “Predicting the pressure” with “Predicting the choke point pressure” in the legend of Figure 8. Also, please replace “plot point” with “point”.

Page 13 – I did not understand what is the “group velocity of a longitudinal small amplitude wave travelling along a cylinder $u = cg$.” Please provide more details so that readers can understand the concept without having to read Ref. [49].

Table 1 – Please include the definition of the variables l , l_0 , h and a in the legend of Table 1.

Page 18 - The tube cross-sectional area was estimated based on a 3D reconstruction of the tube geometry. It would be interesting if the authors included a figure showing a 3D reconstruction of a tube geometry to give the reader a sense of what the 3D shape of the tube looks like and how accurate this approach is. In addition, it would be helpful if the authors discussed the accuracy of estimating the internal cross-sectional area based on the external shape of the tube. Did the authors validate their methods by comparing their results with tube laws reported in the literature?

Page 20 – Please report the functional form of the loss coefficient K_d in the “refined model”. In other words, what is the mathematical formula of the loss coefficient K_d shown in Figure 17 as a “refined model”?

Decision letter (RSOS-201951.R0)

Dear Mr Gregory

On behalf of the Editors, we are pleased to inform you that your Manuscript RSOS-201951 "An Experimental Investigation to Model Wheezing in Lungs" has been accepted for publication in Royal Society Open Science subject to minor revision in accordance with the referees' reports. Please find the referees' comments along with any feedback from the Editors below my signature.

Please submit your revised manuscript and required files (see below) no later than 7 days from today's (ie 18-Jan-2021) date. Note: the ScholarOne system will 'lock' if submission of the revision is attempted 7 or more days after the deadline. If you do not think you will be able to meet this deadline please contact the editorial office immediately.

Kind regards,
Royal Society Open Science Editorial Office
Royal Society Open Science

on behalf of Dr Derek Abbott (Associate Editor) and R. Kerry Rowe (Subject Editor)
openscience@royalsociety.org

Reviewer comments to Author:

Reviewer: 1

Comments to the Author(s)

The manuscript is directed towards a physics-based understanding of the onset of self-excited oscillations in flexible tubes with implications on wheezing, i.e. sounds used for diagnosis of pulmonary diseases. The manuscript is clearly written, of high quality. I consider that it can be accepted after some minor revisions as suggested below.

Q1: One must mention that the analysis of wheezing is almost exclusively focused on the on-set of oscillations and their characterization, without a detailed analysis of the mechanisms for sound production. Will the authors comment on this aspect?

Q2: Which has been the main reason for going beyond the typical values of the geometric parameters found in the upper airways, but in the same time not exploring the entire space for h/d (see also Fig. 1)?

Q3: Can the authors comment on the reason for which the expiration phase have been considered? Does not makes sense to focus on both inspiration & expiration?

Q4: The movies are showing that the wall dynamics are significant, with quite a variation of the tube's minimal cross-sectional shape. Can one comment on the fluid flow characteristics associated with such a scenario and the potential impact on wall dynamics? Moreover, unsteady pressure loads on surfaces may also generate sound.

Q5: How the very rigorous analysis on understanding of onset of oscillations on the Starling resistor (of different topologies and under different conditions) can be used for assessing the lungs and their function (as a series of Starling resistors interconnected). How the developed model for predicting self-excited oscillations in bronchi can be used in the clinical world?

Q6: What one can say about the acoustic sources generating wheezing?

Q7: How one can ensure the transfer of information and knowledge to the physicians so that one can make sure that the developed model (for predicting onset of self-excited oscillations with relevance to wheezing in the lungs) can be used in diagnosis?

Reviewer: 2

Comments to the Author(s)

MAJOR COMMENTS

This manuscript investigates the physical mechanisms associated with wheezing (generation of sounds in the lungs). The authors report sophisticated in vitro experiments and a mathematical model of the fluid dynamics in collapsible tubes. Based on these results, they propose a phenomenological model that describes self-excited oscillations as an "aeroelastic flutter in which two modes of vibration - longitudinal and transverse - couple to give resonance" (Page 16, Line

37). From my knowledge of the literature in this field, this manuscript is a major contribution on this topic in many levels, including the description of novel tube laws that account for longitudinal stretching, very carefully conducted in vitro experiments, and the introduction of new concepts such as that self-excited oscillations are a “resonance between two modes of oscillations: a longitudinal mode (waves travelling up and down the tube) and a transverse mode (opening and closing of the tube)” (Page 13, Line 42).

My only major comment is that Section 8 (Applying the Mechanism to the Lung) lacks important details. The authors need to revise this section to provide more details:

(1) The authors state: “For the geometry of the airways we refer to [18,41,50].” The authors should include a table with the length, diameter, and wall thickness of each generation in their lung model.

(2) What modulus of elasticity (Young’s modulus) was used in the lung model? I imagine that the modulus of elasticity plays a major role given that a lung made of “steel” would not collapse. How was the modulus of elasticity of lung structures estimated?

(3) To create Figure 12, I assume the authors used a 1-dimensional model to estimate the pressure-flow relationship in their lung model, but this is not described. How was the pressure field in the lung model computed?

(4) How was the pc^* computed for each lung generation?

MINOR COMMENTS

I have several suggestions to improve readability.

Page 3, Line 15 – This sentence is a bit strange. Isn’t air always “the fluid flowing in the airways”? I assume that references [5,6] investigated other gases (helium?). But I recommend revising this sentence for better clarity.

Page 3, Line 39 – Elliott and Dawson [1] is another example of experimental work that used air as the working fluid.

[1] – Elliott E.A. and Dawson S.V. (1977) Test of wave-speed theory of flow limitation in elastic tubes. *J. Appl. Physiol.* 43(3), 516-522.

Page 4, Line 34 – Please specify the source of the rubber tubes used in the experiments. Also, please explain if the modulus of elasticity ($E = 1$ MPa) and Poisson’s ratio ($\nu = 0.5$) were measured in your lab or if the mechanical properties of rubber come from the tube manufacturer.

Page 4, Lines 36-45 – As the authors stated, “it is vital for experimentalists to describe, quantitatively, all aspects of their experiment” (Page 3, Line 29). In this spirit, I recommend that the authors report as a supplementary material all the dimensions of their experimental setup, such as the volumes of the settling chambers (3 and 3’ in Fig. 2), the diameters and lengths of the upstream (5) and downstream (5’) tubes, the dimensions of the contraction (6) and expansion (6’), etc.

Page 4, Lines 36-45 – How was the rubber tube clamped to the contraction and expansion? (I assume there was a slight change in diameter as the rubber tube was clamped over the

contraction and expansion.) This information may be useful to provide the boundary conditions in future computational work that may try to reproduce this experiment.

Page 6, Line 18 – The following sentence is unclear because Figure 3 does not display the pressure difference $p_1 - p_2$: “This shows that the onset of oscillation introduces a pressure loss between p_1 and p_2 and restricts the flowrate.” Perhaps the authors can add a line to Figure 3a showing the pressure difference $p_1 - p_2$.

Page 6, Line 37 – Please replace “affect” with “effect”.

Page 6, Line 51 – The authors state that “Videos from our study can be seen in the supplementary materials for this paper.” I could not find supplementary materials on the Royal Society Open Science website. I watched the video corresponding to Figure 4 whose link is Ref. [59], but it is unclear what are the “supplementary materials” the authors are referring to in this sentence.

Figure 3 – The legend of panel (a) in Figure 3 shows a dashed line for p_1 and a solid line for p_2 . However, the plot shows dashed lines for both p_1 and p_2 . I assume that p_2 is the lower border of the solid colored area, but this is not entirely clear given the disconnect with the legend.

Figure 4 – Please explain in the legend of Figure 4 that the white dots on the rubber tube are light dots to improve visualization.

Page 8, Line 27 – I believe the variable “a” is used here for the first time without explaining that “a” is the radius of the undeformed tube.

Page 8, Line 41 – It is unclear why the term $(l/10)$ is included in both the left and right sides of the equation used to define the axes of Figure 5. Wouldn't it be easier to use $A_c/(\pi a^2)$ as the X-axis in Figure 5?

Figure 6 – Please provide in the legend of Figure 6 the tube geometry for which this fluid law refers to.

Page 11, Line 34 – Please replace the subsection title “Mechanism for Onset” with “Mechanism for Onset of Oscillations”. Likewise, please replace “onset” with “onset of oscillations” in line 35.

Figure 8 – Please replace “Predicting the pressure” with “Predicting the choke point pressure” in the legend of Figure 8. Also, please replace “plot point” with “point”.

Page 13 – I did not understand what is the “group velocity of a longitudinal small amplitude wave travelling along a cylinder $u = cg$.” Please provide more details so that readers can understand the concept without having to read Ref. [49].

Table 1 – Please include the definition of the variables $l/10$, h and a in the legend of Table 1.

Page 18 - The tube cross-sectional area was estimated based on a 3D reconstruction of the tube geometry. It would be interesting if the authors included a figure showing a 3D reconstruction of a tube geometry to give the reader a sense of what the 3D shape of the tube looks like and how accurate this approach is. In addition, it would be helpful if the authors discussed the accuracy of estimating the internal cross-sectional area based on the external shape of the tube. Did the authors validate their methods by comparing their results with tube laws reported in the literature?

Page 20 – Please report the functional form of the loss coefficient K_d in the “refined model”. In other words, what is the mathematical formula of the loss coefficient K_d shown in Figure 17 as a “refined model”?

===PREPARING YOUR MANUSCRIPT===

- one version identifying all the changes that have been made (for instance, in coloured highlight, in bold text, or tracked changes);
- a 'clean' version of the new manuscript that incorporates the changes made, but does not highlight them. This version will be used for typesetting.

===PREPARING YOUR REVISION IN SCHOLARONE===

- 1) One version identifying all the changes that have been made (for instance, in coloured highlight, in bold text, or tracked changes);
 - 2) A 'clean' version of the new manuscript that incorporates the changes made, but does not highlight them.
 - An individual file of each figure (EPS or print-quality PDF preferred [either format should be produced directly from original creation package], or original software format).
 - An editable file of each table (.doc, .docx, .xls, .xlsx, or .csv).
 - An editable file of all figure and table captions.
- Note: you may upload the figure, table, and caption files in a single Zip folder.
- Any electronic supplementary material (ESM).
 - If you are requesting a discretionary waiver for the article processing charge, the waiver form must be included at this step.
 - If you are providing image files for potential cover images, please upload these at this step, and inform the editorial office you have done so. You must hold the copyright to any image provided.
 - A copy of your point-by-point response to referees and Editors. This will expedite the preparation of your proof.

- Ensure that your data access statement meets the requirements at <https://royalsociety.org/journals/authors/author-guidelines/#data>. You should ensure that you cite the dataset in your reference list. If you have deposited data etc in the Dryad repository, please only include the 'For publication' link at this stage. You should remove the 'For review' link.
- If you are requesting an article processing charge waiver, you must select the relevant waiver option (if requesting a discretionary waiver, the form should have been uploaded at Step 3 'File upload' above).
- If you have uploaded ESM files, please ensure you follow the guidance at <https://royalsociety.org/journals/authors/author-guidelines/#supplementary-material> to include a suitable title and informative caption. An example of appropriate titling and captioning may be found at https://figshare.com/articles/Table_S2_from_Is_there_a_trade-off_between_peak_performance_and_performance_breadth_across_temperatures_for_aerobic_scope_in_teleost_fishes_/3843624.

Author's Response to Decision Letter for (RSOS-201951.R0)

See Appendix A.

Decision letter (RSOS-201951.R1)

Dear Mr Gregory,

It is a pleasure to accept your manuscript entitled "An Experimental Investigation to Model Wheezing in Lungs" in its current form for publication in Royal Society Open Science.

on behalf of Dr Derek Abbott (Associate Editor) and R. Kerry Rowe (Subject Editor)
openscience@royalsociety.org

Appendix A

Response to referees for Royal Society Open Science for Manuscript RSOS-201951 “An Experimental Investigation to Model Wheezing in Lungs”

Dr Alastair Gregory (alg57@cam.ac.uk)

Dr Anurag Agarwal

Professor Joan Lasenby

18 January 2021

For this response, the reference numbering used corresponds to the first draft submitted to Royal Society Open Science.

Reviewer 1

1. One must mention that the analysis of wheezing is almost exclusively focused on the on-set of oscillations and their characterization, without a detailed analysis of the mechanisms for sound production. Will the authors comment on this aspect?
Yes, we have focussed on onset in this work. We believe that there are probably two main aspects to the sound production: The oscillating tube will produce a varying mass flux through the airway, which will act as a sound source, and the moving walls themselves will act as one two, though due to how the walls move possibly a less efficient one. It is also necessary to understand how the sound “gets out” of the chest. For the unsteady mass flow source, the sound must pass through the wall of the airways and the parenchyma, whereas the wall movements will produce sound waves directly in the parenchyma. Providing a full investigation of this is not the main aim of our work here but we have included some more discussion of it in §8
2. Which has been the main reason for going beyond the typical values of the geometric parameters found in the upper airways, but in the same time not exploring the entire space for h/d (see also Fig. 1)?
We were limited by the kind of rubber tubes we could source, which were obtained by cutting off the ends of rocket balloons. After cutting up the balloons, we tested the elastic moduli of the rubber ourselves with an Instron machine. This has been included in §2.
3. Can the authors comment on the reason for which the expiration phase have been considered? Does not makes sense to focus on both inspiration & expiration?
We focussed on expiration because this is the more common part of the breathing cycle when wheezing occurs (ref [5]), a note of this is now made in §2. It also meant that the experimental setup was simpler, removing the need for a pressure chamber around the tube.
4. The movies are showing that the wall dynamics are significant, with quite a variation of the tube's minimal cross-sectional shape. Can one comment on the fluid flow characteristics associated with such a scenario and the potential impact on wall dynamics? Moreover, unsteady pressure loads on surfaces may also generate sound.
The flow characteristics after the onset of oscillations are going to be very complicated. In this paper, our focus has been on the onset and the conditions

leading to the onset. We are conducting numerical simulations on our setup and that could help us understand the flow-field better. Regarding the generated sound field, the two main sources are the unsteady mass flow going in the rig (an efficient monopole source) and the vibration of the rubber tube. The details of flow-field post-onset and how the sound is generated could be a subject of another paper.

5. How the very rigorous analysis on understanding of onset of oscillations on the Starling resistor (of different topologies and under different conditions) can be used for assessing the lungs and their function (as a series of Starling resistors interconnected). How the developed model for predicting self-excited oscillations in bronchi can be used in the clinical world?

If we have a reliable model that predicts oscillation onset and frequency based on tube geometry and material properties, then the aim is to invert this relationship. If a wheeze is heard, and its source in the lungs can be located, then the frequency could be used to gauge changes in material properties of the lung (e.g. fibrosis). This could also be combined with information on lung flow (measured with PEFR or similar) to gauge at what flow rates the wheezing was happening, which would give a measure of the health of the patient.

6. What one can say about the acoustic sources generating wheezing?

See the discussion for question 1.

7. How one can ensure the transfer of information and knowledge to the physicians so that one can make sure that the developed model (for predicting onset of self-excited oscillations with relevance to wheezing in the lungs) can be used in diagnosis?

To cross the bridge to clinical practice more in-vivo results would be needed. We have collected data on breathing sounds from patients as part of a related project so this is a good avenue for further work. We are also working on a University spinout to commercialise our work to ensure clinical deployment (biophonics.co.uk).

Reviewer 2

Major Comments

1. The authors state: "For the geometry of the airways we refer to [18,41,50]." The authors should include a table with the length, diameter, and wall thickness of each generation in their lung model.
We have created a new appendix (C) describing the Horsefield model (refs [41,50]) for the human lung in more detail (including the table), which is what we used. In addition, Ref [18] allows us to estimate the axial strain of the tubes in the lung.
2. What modulus of elasticity (Young's modulus) was used in the lung model? I imagine that the modulus of elasticity plays a major role given that a lung made of "steel" would not collapse. How was the modulus of elasticity of lung structures estimated?

We used a Young's modulus of 1MPa and assumed the material was incompressible. The incompressibility and approximate elasticity of soft tissue comes from several sources¹²³⁴.

3. To create Figure 12, I assume the authors used a 1-dimensional model to estimate the pressure-flow relationship in their lung model, but this is not described. How was the pressure field in the lung model computed?

We actually just take $p_c = -20$ cmH₂O everywhere, as this gives an upper bound on what the real $-p_c$ can be. Figure 12 shows that even when we used this over-estimate p_c/p_c^* is still less than 0.5. We have done work where we used a 1D model, based on the Darcy-Weisbach equation⁵ but we did not find that this was needed for the conclusions in this paper, so it was not included for simplicity. For this analysis see Gregory AL. 2019 A Theory for Wheezing in Lungs (Doctoral thesis). (doi:<https://doi.org/10.17863/CAM.32286>). We have made this clearer in §8.

4. How was the p_c^* computed for each lung generation?

It is calculated using eqn 4.1b, using geometry from the Horsefield model and material properties according to references given in the answer to question 2. This has been clarified in §8.

Minor Comments

1. Page 3, Line 15 – This sentence is a bit strange. Isn't air always "the fluid flowing in the airways"? I assume that references [5,6] investigated other gases (helium?). But I recommend revising this sentence for better clarity.

This interpretation is right, we have revised the sentence.

2. Page 3, Line 39 – Elliott and Dawson⁶ is another example of experimental work that used air as the working fluid.

Thanks. We have included discussion of it in §1.

3. Page 4, Line 34 – Please specify the source of the rubber tubes used in the experiments. Also, please explain if the modulus of elasticity ($E = 1$ MPa) and Poisson's ratio ($\nu = 0.5$) were measured in your lab or if the mechanical properties of rubber come from the tube manufacturer.

This has been done.

4. Page 4, Lines 36-45 – As the authors stated, "it is vital for experimentalists to describe, quantitatively, all aspects of their experiment" (Page 3, Line 29). In this spirit, I recommend that the authors report as a supplementary material all the dimensions of their experimental setup, such as the volumes of the settling

¹ Chen EJ, Novakofski J, Kenneth Jenkins W, O'Brien WD. 1996 Young's modulus measurements of soft tissues with application to elasticity imaging. *IEEE Trans. Ultrason. Ferroelectr. Freq. Control* **43**, 191–194. (doi:10.1109/58.484478)

² Fung YC, Cowin SC. 1993 *Biomechanics: Mechanical Properties of Living Tissues*. 2nd edn. Springer.

³ Wang JY, Mesquida P, Lee TH. 2011 Young's modulus measurement on pig trachea and bronchial airways. In *Proceedings of the Annual International Conference of the IEEE EMBS*, pp. 2089–2092. Boston, MA. (doi:10.1109/IEMBS.2011.6090388)

⁴ Wang JY, Mesquida P, Pallai P, Corrigan CJ, Lee TH. 2011 Dynamic elastic properties of human bronchial airway tissues. *arXiv*

⁵ Jarić M, Kolendić P, Budimir N, Genić V. 2011 A Review of Explicit Approximations of Colebrook's Equation. *FME Trans.* **39**, 67–71.

⁶ Elliott E.A. and Dawson S.V. (1977) Test of wave-speed theory of flow limitation in elastic tubes. *J. Appl. Physiol.* **43**(3), 516–522.

chambers (3 and 3' in Fig. 2), the diameters and lengths of the upstream (5) and downstream (5') tubes, the dimensions of the contraction (6) and expansion (6'), etc. **A more thorough description of the setup can be seen in a thesis (free to download)⁷ including the dimensions of the parts discussed here. We have now cited this thesis in §2.**

5. Page 4, Lines 36-45 – How was the rubber tube clamped to the contraction and expansion? (I assume there was a slight change in diameter as the rubber tube was clamped over the contraction and expansion.) This information may be useful to provide the boundary conditions in future computational work that may try to reproduce this experiment.
We 3D printed mounts that matched the inner diameter of the rubber tubes, and had a conical outer shape so that by pulling the rubber tubes onto the cones, taping them there, and being careful not to twist them, we could get repeatable results. More detail is given in the thesis, which is now cited.
6. Page 6, Line 18 – The following sentence is unclear because Figure 3 does not display the pressure difference $p_1 - p_2$: “This shows that the onset of oscillation introduces a pressure loss between p_1 and p_2 and restricts the flowrate.” Perhaps the authors can add a line to Figure 3a showing the pressure difference $p_1 - p_2$.
The shaded region between p_1 and p_2 indicates the pressure difference. We have amended the line to clarify this.
7. Page 6, Line 37 – Please replace “affect” with “effect”.
This has been done.
8. Page 6, Line 51 – The authors state that “Videos from our study can be seen in the supplementary materials for this paper.” I could not find supplementary materials on the Royal Society Open Science website. I watched the video corresponding to Figure 4 whose link is Ref. [59], but it is unclear what are the “supplementary materials” the authors are referring to in this sentence.
The supplementary material is a dataset. We have cited this directly now.
9. Figure 3 – The legend of panel (a) in Figure 3 shows a dashed line for p_1 and a solid line for p_2 . However, the plot shows dashed lines for both p_1 and p_2 . I assume that p_2 is the lower border of the solid coloured area, but this is not entirely clear given the disconnect with the legend.
 p_2 is the bottom of the coloured area, which does have a solid line.
10. Figure 4 – Please explain in the legend of Figure 4 that the white dots on the rubber tube are light dots to improve visualization.
This had been done.
11. Page 8, Line 27 – I believe the variable “ a ” is used here for the first time without explaining that “ a ” is the radius of the undeformed tube.
 a has now been defined in §2.
12. Page 8, Line 41 – It is unclear why the term (l/l_0) is included in both the left and right sides of the equation used to define the axes of Figure 5. Wouldn't it be easier to use $A_c/(\pi \cdot a^2)$ as the X-axis in Figure 5?
The cross-sectional of the tube at its centre gets smaller when it is tensioned axially (even when $p_c=0$), so if $A_c/(\pi \cdot a^2)$ is used as the x-axis, the curves do not collapse onto each other properly. This has been explained further in the paper.

⁷ Gregory AL. 2019 A Theory for Wheezing in Lungs (Doctoral thesis). (doi:<https://doi.org/10.17863/CAM.32286>).

13. Figure 6 – Please provide in the legend of Figure 6 the tube geometry for which this fluid law refers to.

This has been done.

14. Page 11, Line 34 – Please replace the subsection title “Mechanism for Onset” with “Mechanism for Onset of Oscillations”. Likewise, please replace “onset” with “onset of oscillations” in line 35.

This has been done.

15. Figure 8 – Please replace “Predicting the pressure” with “Predicting the choke point pressure” in the legend of Figure 8. Also, please replace “plot point” with “point”.

This has been done.

16. Page 13 – I did not understand what is the “group velocity of a longitudinal small amplitude wave travelling along a cylinder $u = cg$.” Please provide more details so that readers can understand the concept without having to read Ref. [49].

We have expanded the explanation of the group velocity derivation.

17. Table 1 – Please include the definition of the variables l/l_0 , h and a in the legend of Table 1.

This has been done.

18. Page 18 - The tube cross-sectional area was estimated based on a 3D reconstruction of the tube geometry. It would be interesting if the authors included a figure showing a 3D reconstruction of a tube geometry to give the reader a sense of what the 3D shape of the tube looks like and how accurate this approach is. In addition, it would be helpful if the authors discussed the accuracy of estimating the internal cross-sectional area based on the external shape of the tube. Did the authors validate their methods by comparing their results with tube laws reported in the literature?

We have expanded on the description of the tube cross-section reconstruction in the paper. We have included information about how uncertainty is estimated. We have found comparison with other tube laws difficult, since the shape of the tube law seems to fundamentally change when significant axial tension is included. Specifically, the shape of the tube law becomes much smoother with significant axial tension. Our results do not display the abrupt collapses in tube area that is seen in other tube laws. We would infer that there is a transition between the abrupt behaviour seen by others and the smoother behaviour we see, but it seems that this transition is itself very abrupt. We have been performing numerical simulations to obtain the tube laws and see good agreement between the numerical and experimental results.

19. Page 20 – Please report the functional form of the loss coefficient K_d in the “refined model”. In other words, what is the mathematical formula of the loss coefficient K_d shown in Figure 17 as a “refined model”?

The refined model is a piecewise cubic interpolation of chosen points (given in the paper).